# A functional family of fluorescent nucleotide analogues to investigate actin dynamics and energetics

Jessica Colombo[1,3], Adrien Antkowiak [1,3], Konstantin Kogan [2], Tommi Kotila [2], Jenna Elliott [1], Audrey Guillotin[1], Pekka Lappalainen [2] & Alphée Michelot [1✉]

Actin polymerization provides force for vital processes of the eukaryotic cell, but our understanding of actin dynamics and energetics remains limited due to the lack of high-quality probes. Most current probes affect dynamics of actin or its interactions with actin-binding proteins (ABPs), and cannot track the bound nucleotide. Here, we identify a family of highly sensitive fluorescent nucleotide analogues structurally compatible with actin. We demonstrate that these fluorescent nucleotides bind to actin, maintain functional interactions with a number of essential ABPs, are hydrolyzed within actin filaments, and provide energy to power actin-based processes. These probes also enable monitoring actin assembly and nucleotide exchange with single-molecule microscopy and fluorescence anisotropy kinetics, therefore providing robust and highly versatile tools to study actin dynamics and functions of ABPs.

---

[1] Aix Marseille Univ, CNRS, IBDM, Turing Centre for Living Systems, 13288 Marseille, France. [2] HiLIFE Institute of Biotechnology, P.O. Box 56, University of Helsinki, 00014 Helsinki, Finland. [3] These authors contributed equally: Jessica Colombo, Adrien Antkowiak. ✉email: alphee.michelot@univ-amu.fr

The actin cytoskeleton is a force-generating machinery involved in many processes of eukaryotic cells, such as migration, morphogenesis, cytokinesis, and endocytosis[1]. Actin filaments within actin networks assemble from a pool of globular actin monomers (G-actin), and the polymerization reaction against cellular membranes is one of the physiological mechanisms by which actin networks exert forces[2–4]. A general characteristic of actin-based force generation is that it is powered by adenosine triphosphate (ATP). As an ATPase, which binds to ATP with a 1:1 stoichiometry, actin transduces the energy liberated from the hydrolysis of ATP into mechanical work. It is assumed that at steady state, actin filaments assemble at their barbed ends from a pool of G-actin bound to ATP. Subsequently, ATP hydrolysis and phosphate release occur within actin filaments, and filaments disassemble from their pointed ends in the ADP state[5]. The state of the bound nucleotide also triggers conformational changes in actin, thereby modulating the affinity of actin for a number of actin-binding proteins (ABPs)[6]. Recharging of actin monomers with ATP provides the chemical energy that is required to sustain a dynamic actin turnover in the cell, and fluctuations of cellular energetic levels are correlated with important behavioral changes[7,8].

Because the spontaneous turnover of actin filaments is slow, cells employ a plethora of ABPs to catalyze every step of this cycle[6]. For instance, the Arp2/3 complex accelerates nucleation of new actin filaments from the sides of pre-existing mother filaments, whereas heterodimeric capping protein controls the elongation of actin filaments at their rapidly growing barbed ends[6]. Other proteins such as actin-depolymerizing factor (ADF)/ cofilin sever actin filaments to enhance their disassembly[9,10], whereas profilin and cyclase-associated protein accelerate the nucleotide exchange on the resulting ADP actin monomers, thereby contributing in replenishing the cellular pool of ATP-G-actin[11–13]. Moreover, actin filaments can be decorated by various proteins, such as tropomyosins, which affect the stability of filaments and modulate their interactions with other proteins[14,15]. Uncovering the molecular mechanisms by which different ABPs affect actin is crucial for understanding how ATP is consumed by cellular actin networks to generate force.

During recent years, significant progress has been made in the field through the development of in vitro approaches for visualizing the dynamics of individual actin filaments[16,17], as well as in vitro reconstituted systems to study actin-based force generation[10,18]. These approaches generally require the visualization of actin filaments, either by specific actin-binding molecules or covalent labeling of actin. However, these labeling approaches have limitations. Fluorescent actin probes, such as phalloidin and LifeAct affect the dynamics of actin filaments and do not uniformly label all actin filament structures[19–21]. On the other hand, chemical labeling does not work robustly on all actins[22] and may affect the binding of some key ABPs to G-actin[11] or to actin filaments[14]. Thus there is a need for robust approaches for visualizing actin filaments.

Moreover, our understanding of actin recycling has suffered from technical difficulties in following the state of the nucleotide bound to actin, both in vivo and in vitro. As for most other ATPases, there is a lack of good tools and experimental approaches to follow the dynamics of the bound nucleotides. Most of our current knowledge is based on in vitro experiments using $N^6$-etheno-ATP (εATP; Fig. 1a), which was discovered nearly 50 years ago and whose binding to actin can be monitored by a fluorescence increase at 338 nm[23]. While εATP has been historically important to understand these mechanisms, it lacks sensitivity and remains a poor probe for fluorescence microscopy[24].

Here, we identify a family of fluorescent nucleotides, $N^6$-(6-Amino)hexyl-ATP derivatives, as a robust tool for studying actin

dynamics. We provide evidence that these molecules are functional with actin and do not inhibit the activities of key ABPs on actin. These fluorescent nucleotides therefore represent a powerful class of actin probes, since they are detectable down to the single molecule level and allow tracking of actin assembly in vitro without modifying the actin itself or adding labeled proteins. We also describe fluorimetry methods, which enable quantification of the dynamics of the nucleotide bound state during actin network formation and turnover.

## Results

**Identification of a family of fluorescent nucleotide analogs interacting with actin.** We hypothesized that directly labeling the actin-bound nucleotide with fluorescent dyes would offer a powerful way to track simultaneously actin assembly and the dynamics of the bound nucleotide. The problem with this strategy is that dyes are larger than the nucleotides themselves, and usually prevent their correct insertion within the nucleotide-binding pockets of proteins. Different chemistries are nevertheless possible, where bright dyes are covalently linked to the base, ribose, or phosphate groups of ATP[25] (Fig. 1a). We first tested a $2'(3')$-$O$-(N-(2-(amino)ethyl)carbamoyl) ATP (EDA-ATP) analog, EDA-ATP-ATTO-488, because such chemistry was previously used in studies of myosin subfragment heavy meromyosin[26,27]. We incubated EDA-ATP-ATTO-488 with G-actin to evaluate its ability to exchange with the ATP bound to G-actin. As fluorescent ATP analogs are relatively small (MW < 1.5 kDa) compared to actin (MW = 42 kDa), and because ATTO-488 has a relatively long fluorescence lifetime ($\tau_{fl} = 4.1$ ns), we expected that binding could be detected by an increase of the fluorescence anisotropy signal. However, anisotropy signals before and after a 30-min incubation were similar, suggesting a lack of binding. Moreover, a phosphate modified ATP nucleotide analog, γ-[6-Aminohexyl]-ATP-ATTO-488, did not display detectable binding to G-actin based on this assay (Fig. 1b). Structural analysis of the positions of these modifications in actin's ATP-binding pocket reveals steric clashes for dyes linked to the ribose or phosphate groups of ATP with the interior of the actin molecule, thus providing a structural explanation for their inability to bind actin (Fig. 1c). However, the actin structure suggests that modifications of the base of ATP might be less detrimental for an interaction with actin, as such conjugations are expected to orient the dyes away from the interior of actin (Fig. 1c, d). We therefore incubated a third ATP analog, $N^6$-(6-Amino)hexyl-ATP-ATTO-488 (referred to in the subsequent parts of the article as ATP-ATTO-488), with G-actin. It triggered a clear increase of fluorescence anisotropy, demonstrating that this ATP analog is capable of binding to actin (Fig. 1b).

The dye in ATP-ATTO-488 is conjugated to ATP via a relatively long linker, which is likely to be flexible and may allow multiple orientations for the dye in respect to actin (Fig. 1d). To further study the mechanism by which ATP-ATTO-488 interacts with actin, we co-crystallized ATP-ATTO-488 actin monomer in complex with an actin-depolymerizing factor homology domain of twinfilin (Supplementary Fig. 1a, b), and solved the structure by molecular replacement (Supplementary Table 1). The obtained structure (Fig. 1e) was almost identical (RMSD = 0.451) to the corresponding complex determined with ATP-bound actin[28]. Despite relatively good resolution (2.56 Å) and sufficient space in the crystal to accommodate the ATTO-488 dye (Supplementary Fig. 1c), we could reliably model neither the linker nor the dye itself, supporting the idea that the linker is highly flexible, and therefore the dye is not identifiable by X-ray in the crystal. However, we observed some additional electron density in the vicinity of ATP at the distance where the ATTO-488 dye might

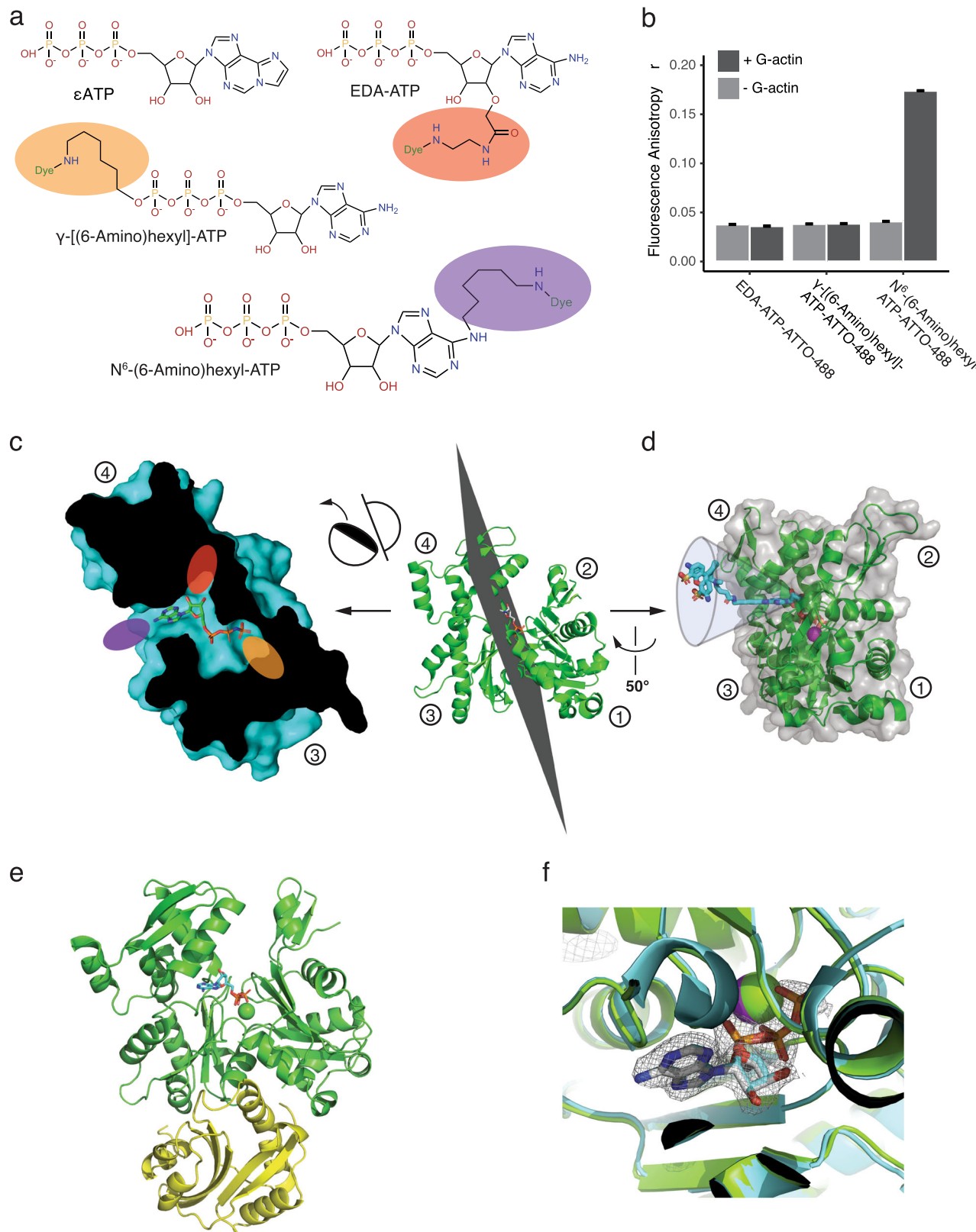

reside (Supplementary Fig. 1d). Importantly, electron density for the ATP part of the ATP-ATTO-488 molecule was clearly visible and resided in an almost identical position as compared to ATP in the previous structure (Fig. 1f). Together, these data show that the ATP part of ATP-ATTO-488 binds to actin in a very similar manner to unlabeled ATP, and that the dye conjugated to ATP through flexible linker is positioned away from the nucleotide-binding pocket of actin.

In the next sections, we provide evidence that $N^6$-(6-Amino) hexyl-ATP derivatives are functional with actin and do not inhibit the activities of key ABPs on actin. These fluorescent nucleotides therefore represent a powerful class of actin

**Fig. 1 Comparison of the binding properties of different families of fluorescent ATP analogs to monomeric actin (G-actin) by steady-state fluorescence anisotropy.** ATP adenosine triphosphate, εATP N[6]-etheno-ATP, EDA-ATP 2′(3′)-O-(N-(2-(amino)ethyl)carbamoyl)-ATP. **a** Skeletal formulas of the families of fluorescent nucleotides tested in this study. Purple, orange, and red ovals indicate positions of the fluorophore. **b** Binding ability of different ATTO-488 fluorescent ATP analogs (0.2 μM) to G-actin (2 μM) in NFG + MEI buffer, revealed by measuring changes in steady-state anisotropy values 30 min after the initiation of the experiment. Light (resp. dark) gray are values obtained in the absence (resp. presence) of G-actin. Bar graphs indicate mean values and standard deviations. $n = 20$ for each condition. **c** Cut-through view of an actin monomer (3DAW, chain A) with bound ATP. Purple, orange, and red ovals represent the positions of the modifications in the ATP molecules shown in panel **a**. In the case of EDA-ATP (red) and γ-[(6-Amino) hexyl]-ATP (orange), the fluorescent modifications are conjugated to ATP in a way that they point towards the interior of actin, leading to a probable steric clash, whereas in the case of N[6]-(6-Amino)hexyl-ATP, the fluorescent modification (purple) is expected to stick out from the nucleotide-binding pocket of actin. **d** One of the theoretically possible configurations of the N[6]-(6-Amino)hexyl-ATP molecule (cyan), when bound to G-actin (green). Due to the flexible linker between the ATP moiety and the dye, the molecule is expected to obtain multiple different orientations (represented by a funnel). Circled numbers mark the actin subdomains. **e** Crystal structure of the complex of rabbit muscle N[6]-(6-Amino)hexyl-ATP-actin (green) and an ADF-H domain from mouse twinfilin (yellow). **f** Comparison of the positions of nucleotide moieties of ATP-ATTO-488 (gray sticks) in actin (green cartoon) from the crystal structure (6YP9) with the crystal structure of same proteins (3DAW) in the presence of unlabeled ATP (cyan sticks) and actin (cyan cartoon). The sphere represents the metal ion bound to the nucleotide (magenta – this structure, green – 3DAW). Please note that the protein structures and the positions of nucleotides are nearly identical in both cases. Source data are provided as a Source Data file.

probes, since they are detectable down to the single molecule level and allow tracking of actin assembly in vitro without modifying the actin itself or adding labeled proteins. We also describe fluorimetry methods, which enable quantification of the dynamics of the nucleotide bound state during actin network formation and turnover.

**Fluorescence anisotropy kinetics as a sensitive method to characterize dynamics of nucleotide exchange on actin monomers.** We next characterized more precisely the binding properties of ATP-ATTO-488 to G-actin, and evaluated the sensitivity of these fluorophores for future experimentations. For this purpose, we compared the exchange kinetics of εATP and ATP-ATTO-488 with G-actin by following changes of fluorescence intensities or fluorescence anisotropies over time. Normalizing both curves to equivalent plateaus indicated that ATP-ATTO-488′s and εATP's exchange dynamics are slightly different from each other (Fig. 2a). Importantly, the fluorescence intensity of ATP-ATTO-488 remained constant as binding occurred (Supplementary Fig. 2a), time scales of the exchange kinetics with ATP-ATTO-488 were comparable at any concentration (Supplementary Fig. 2b), and binding to actin could generally be detected down to 20 pM (Fig. 2b). These results indicate a much higher sensitivity of ATP-ATTO-488 compared to εATP, over at least four orders of magnitude, as nucleotide exchange with εATP cannot be recorded precisely below ~0.2 μM (Supplementary Fig. 2c). Another advantage of fluorescence anisotropy is that it measures the apparent mobility of the ATP-ATTO-488 molecules, is not sensitive to potential photobleaching, and thus allows one to measure meaningful signals over long periods of time.

We next compared the actin-binding properties of ATP-ATTO-488 and ATP. Extreme values of anisotropy $r_{MIN}$ and $r_{MAX}$ represent unbound or fully bound states of ATP-ATTO-488. They can be evaluated by measuring anisotropy values when ATP-ATTO-488 is free in solution or fully bound to actin, respectively. These values enabled us to interpret intermediate values of anisotropy first as binding ratios, and then as concentrations of bound or unbound nucleotides (Fig. 2c and Supplementary Fig. 2d, e). We first performed experiments in which ATP-ATTO-488 was added to ATP-G-actin. In this condition, the main reaction controlling the rate is the dissociation of ATP from actin monomers $k_{-,ATP}$ (Fig. 2d). Then, we performed experiments where an excess of ATP was added. In this case, the main reaction controlling the rate of the reaction is the dissociation of ATP-ATTO-488 from actin monomers $k_{-,ATP-ATTO-488}$ (Fig. 2e). Kinetic data were eventually fitted together with a single set of kinetic parameters to obtain rate

constants. We obtained values of $k_{-,ATP} \approx (2.8 \pm 0.3) \times 10^{-3} \, s^{-1}$ and $k_{-,ATP-ATTO-488} \approx (1 \pm 0.4) \times 10^{-3} \, s^{-1}$. Our model is not sensitive to the value of each association constant, but is sensitive to their ratio $k_{+,ATP}/k_{+,ATP-ATTO-488} \approx 1.2 \pm 0.05$ (Supplementary Table 2). The sensitivity of this method also permits the comparison of exchange kinetics precisely in various experimental conditions. For instance, while addition of 50 mM KCl at high concentration of G-actin triggers actin filament assembly at similar time-scales as nucleotide exchange, experiments performed at lower concentrations of G-actin ensure that nucleotide exchange is recorded before any significant amount of actin has polymerized (Fig. 2f).

Together, these experiments provide evidence that ATP-ATTO-488 displays similar exchange kinetics on actin monomers compared to ATP. However, its dissociation rate from actin is approximately three times slower, and this may be due to weak interactions of the fluorescent dye with actin. Importantly, these data demonstrate that recording ATP-ATTO-488 fluorescence anisotropy kinetics represents a highly sensitive approach for studying nucleotide exchange on actin monomers under physiological conditions, and possibly below the critical concentration of actin assembly.

**Fluorescent nucleotide analogs facilitate single molecule imaging of actin dynamics.** We next applied Total Internal Reflection Fluorescence Microscopy (TIRFM) to reveal whether individual fluorescent ATP molecules can be detected when bound to actin monomers or filaments, and if these ATP analogs could thus be applied as a tool for visualizing actin. N[6]-(6-Amino)hexyl-ATP derivatives are available commercially with a large palette of fluorescent dyes that are specifically developed for imaging. We incubated a 10-fold excess of actin monomers labeled with Alexa-568 to ATP-ATTO-488 to allow nucleotide exchange. The sample was then diluted, placed on a clean coverslip and imaged by TIRFM. Dilution and imaging were performed within a few minutes to limit ATP-ATTO-488 dissociation from G-actin. We did not expect a precise colocalization because only a fraction of actin monomers is labeled with Alexa-568, and only a fraction of actin monomers has exchanged ATP for ATP-ATTO-488. Nevertheless, we observed a clear colocalization of some actin monomers with ATP-ATTO-488 molecules (Fig. 3a).

Importantly, when actin polymerization was triggered simultaneously to the addition of ATP-ATTO-488, we observed a progressive integration of ATP-ATTO-488-bound actin subunits to actin filaments (Fig. 3b and Supplementary Movie 1). This occurs because nucleotides exchange on the actin monomers in

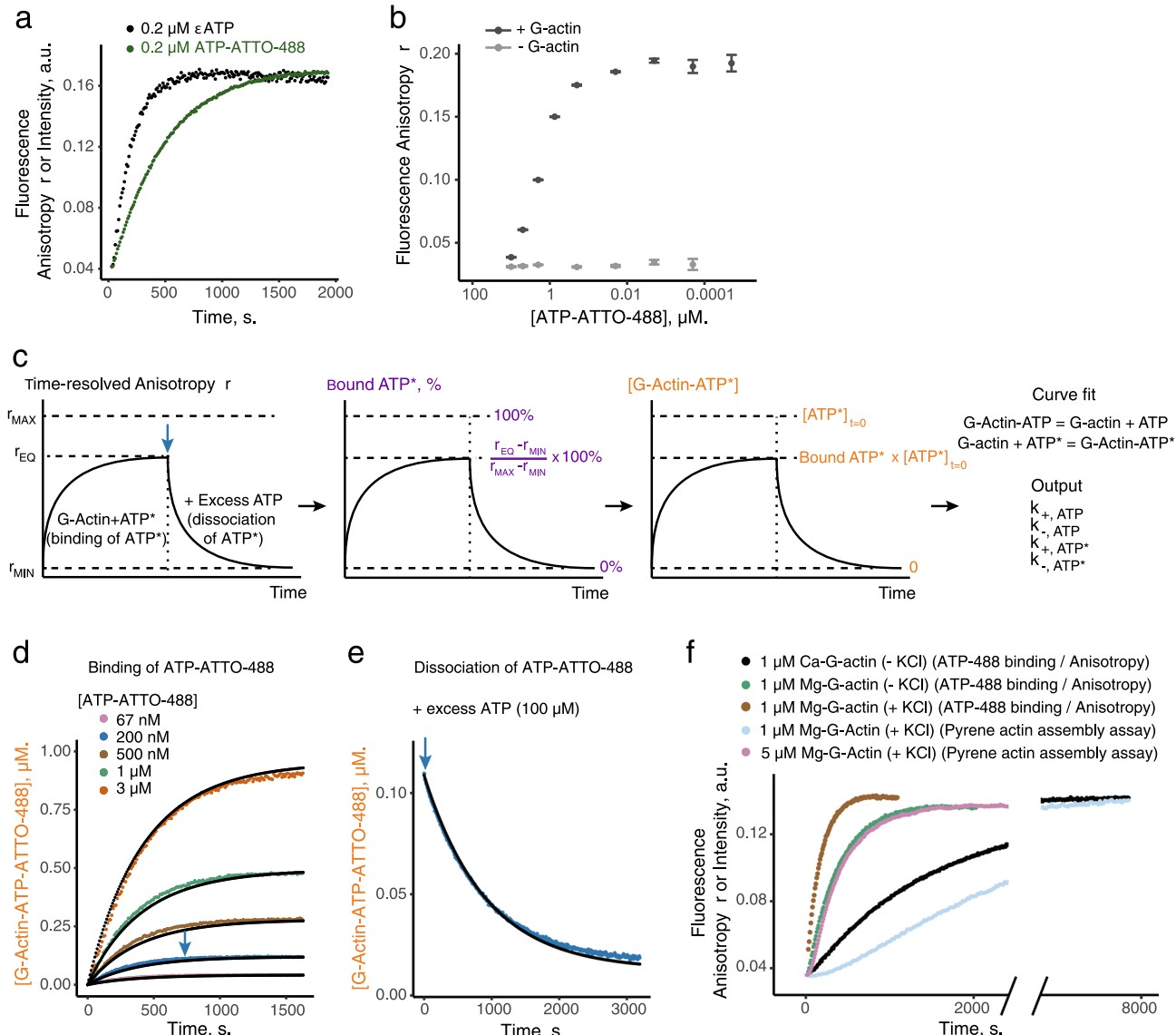

**Fig. 2 A methodology for characterizing the binding kinetics of ATP-ATTO-488 to monomeric actin (G-actin) by fluorescence anisotropy.** ATP adenosine triphosphate, εATP N[6]-etheno-ATP, $r_{MIN}$ free ATP-ATTO-488 anisotropy value, $r_{MAX}$ ATP-ATTO-488 anisotropy value when bound to G-actin, $r_{EQ}$ ATP-ATTO-488 anisotropy value at equilibrium, ATP* is for fluorescent ATP. **a** Comparison of the binding kinetics of εATP (0.2 μM; black curve) or ATP-ATTO-488 (0.2 μM; green curve) to G-actin (2 μM) in NFG + MEI buffer. For εATP, time course of nucleotide exchange was followed by recording the fluorescence signal over time at 10 s time-intervals. For ATP-ATTO-488, time course of nucleotide exchange was followed by measuring changes in the anisotropy value $r$ over time every 10 s. Please note that at plateau the values of the εATP curve decreases due to photobleaching. **b** Sensitivity of the method, evaluated by measuring average steady-state anisotropy values (measured 30 min after initiation of the experiment in NFG + MEI buffer) and standard deviations after exchange of variable concentrations of ATP-ATTO-488 on G-actin (2 μM) indicate that binding to G-actin can be precisely detected down to 20 pM of ATP-ATTO-488. Light (resp. dark) gray represent conditions in the absence (resp. presence) of G-actin. $n = 20$ for each condition. **c** Cartoon representing how kinetic data obtained from ATP* binding or dissociation curves were used to obtain the association and dissociation rates of ATP and ATP* to G-actin. Blue arrow represents the moment when an excess of ATP is added to dissociate ATP*. Y axis colors indicate conversions of anisotropy values (black) into ATP* binding percentage (purple) and concentration of G-actin bound to ATP* (orange). **d** Binding kinetics (colored data) and fit curves (black) in the presence G-actin (2 μM) and increasing concentration of ATP-ATTO 488 in NFG + MEI buffer. Blue arrow indicates the time when an excess of ATP was added in **e**. **e** Kinetics (blue data from the 200-nM ATP-ATTO-488 condition of **d**) and fit curve (black) of ATP-ATTO-488 dissociation from G-actin in the presence of an excess of ATP (100 μM) in NFG + MEI buffer. **d**, **e** were fitted with the same kinetic parameters: $k_{+,ATP} = 10\ \mu M^{-1}\ s^{-1}$; $k_{+,ATP-ATTO-488} = 12\ \mu M^{-1}\ s^{-1}$; $k_{-,ATP} = 2.8 \times 10^{-3}\ s^{-1}$; $k_{-,ATP-ATTO-488} = 10^{-3}\ s^{-1}$. **f** Effect of bound divalent cation ($Ca^{2+}$; in NFG buffer, black curve; 5-fold slower than with $Mg^{2+}$, green curve) and salt (50 mM KCl; in NFG + KMEI buffer, brown curve; 2-fold faster than without KCl) on the binding kinetics of ATP-ATTO-488 (0.2 μM) to G-actin (1 μM). Blue and pink curves are from pyrene-actin assembly assay in NFG + KMEI buffer, normalized to their plateau values, to demonstrate that ATP-ATTO-488 exchange kinetics in the presence of 50 mM KCl can be determined at low concentrations of G-actin for which actin assembly remains negligible at the time scale of nucleotide exchange. Source data are provided as a Source Data file.

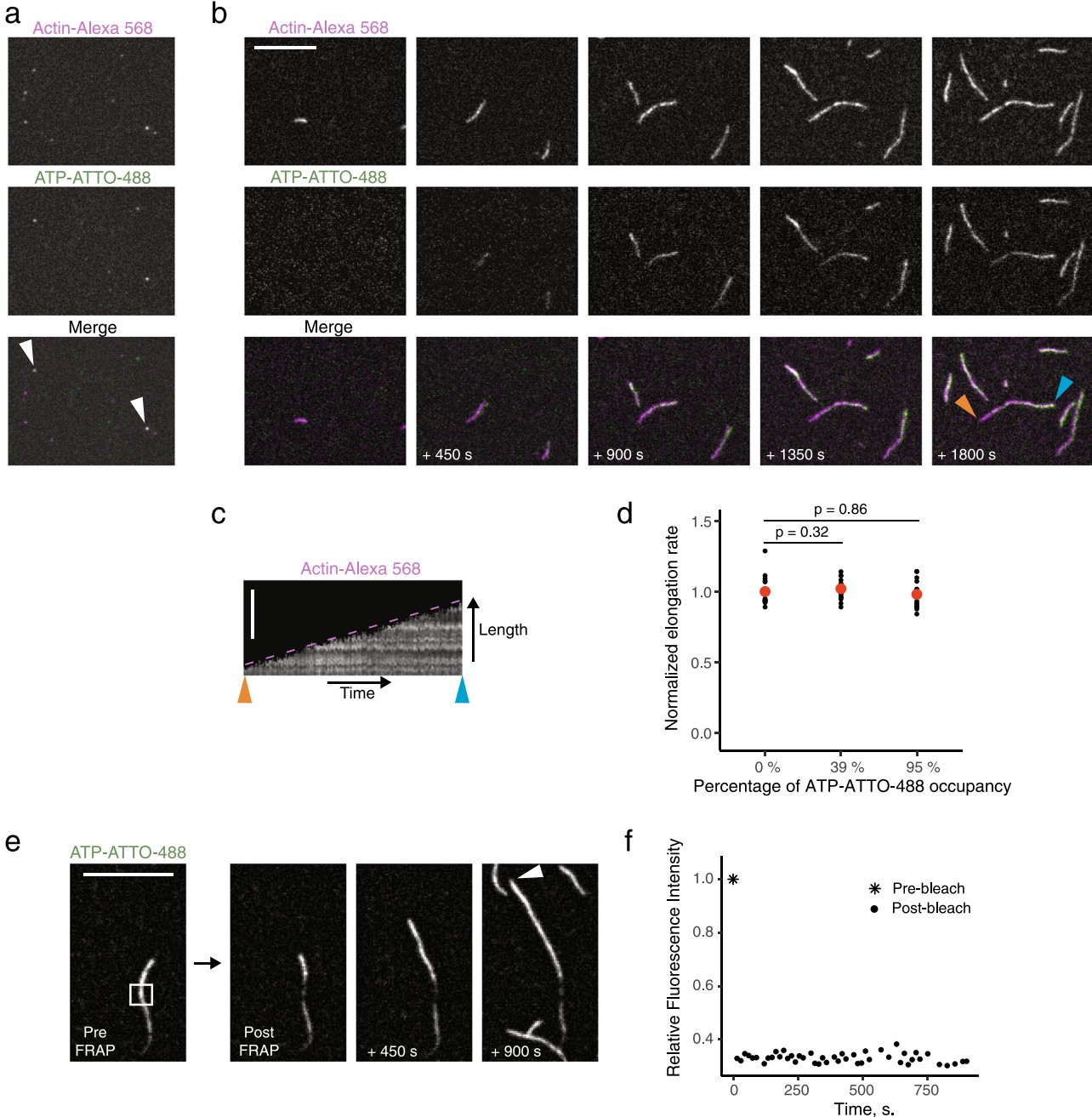

**Fig. 3 Single-molecule imaging of ATP-ATTO-488 bound to G-actin and actin filaments by total internal fluorescence microscopy (TIRFM).** ATP adenosine triphosphate, FRAP fluorescence recovery after photobleaching. Scale bars: 10 μm. **a** Single-molecule imaging of individual ATP-ATTO-488 molecules (green) and Alexa-568-labeled G-actin (magenta). Prior to imaging, ATP-ATTO-488 (0.2 μM) and G-actin (2 μM; 7.5% labeled) were pre-incubated for exchange for 15 min in NFG + MEI buffer before a 10⁵-fold dilution in imaging buffer 1 to isolate single molecules. White arrowheads highlight visible binding. This experiment was repeated independently three times. **b** Time-course of a simultaneous exchange of ATP-ATTO-488 (0.25 μM; green) and polymerization of actin (0.4 μM; 7.5% Alexa-568-labeled; magenta) in the presence of profilin (0.8 μM) in imaging buffer 2. Blue and orange arrowheads indicate the barbed and pointed ends, respectively, of an actin filament, which assembled exclusively from ATP-G-actin monomers at early timepoints, before integrating progressively increasing numbers of ATP-ATTO-488-bound actin subunits. This experiment was repeated independently three times. **c** Filament length vs. time kymograph of the filament indicated with blue and orange arrowheads in **b**. Dotted line shows that slope appears constant over time. **d** Elongation rates of actin filaments polymerized from actin monomers bound to 39% or 95% of ATP-ATTO-488, relative to their elongation rates when bound fully to ATP, indicating that actin subunits bound to ATP-ATTO-488 do not visibly impact actin polymerization. 0, 5 μM, or 125 μM of ATP-ATTO-488 were pre-incubated with 5 μM actin (10% Alexa-568-labeled) for 2 h at room temperature in NFG buffer to reach respectively 0%, 39%, or 95% occupancy. Polymerization was then induced between slides and coverslips by 5-fold dilution in imaging buffer 2 supplemented with 3 μM profilin. Red dots indicate average values. $n = 20$. Statistical significance is given with $p$-values from one-factor ANOVA tests. **e** Photobleaching of a segment of an actin filament labeled with ATP-ATTO-488 demonstrating absence of fluorescence recovery within 900 s, providing evidence that nucleotide exchange does not occur in filamentous form of actin. White arrowhead indicates the growing barbed end. This experiment was repeated independently three times. **f** Quantification of **d**. Source data are provided as a Source Data file.

solution during the first minutes of the experiment (Fig. 2). The fluorescence intensity of ATP-ATTO-488 was independent of Alexa-568 labeling (Supplementary Fig. 3), and the elongation rate of the filaments did not change from the beginning to the end of the experiment (Fig. 3c). Experiments performed at higher occupancy of ATP-ATTO-488 further confirmed that the replacement of ATP for ATP-ATTO-488 does not impact actin polymerization under these conditions (Fig. 3d). We also performed a fluorescence-recovery-after-photobleaching (FRAP) experiment to test whether nucleotide exchange also occurs at the side of actin filaments, because this had never been confirmed visually. No recovery of fluorescence signal was detected within a 15-min period in those actin filament segments where the ATP-ATTO-488 molecules were photobleached (Fig. 3e, f and Supplementary Movie 2), proving that nucleotide exchange cannot occur at the side of actin filaments. Collectively, these data demonstrate that ATP-ATTO-488 provides a robust tool for visualizing actin monomers and filaments in vitro, and that ATP-ATTO-488 does not affect the polymerization rate of actin filaments.

**Fluorescent nucleotide analogues do not interfere with the binding of several essential ABPs**. Beyond binding to actin, we wanted to verify that the binding and activity of several important ABPs was not inhibited by the presence of ATP-ATTO-488.

We first worked with profilin, which catalyzes nucleotide exchange on actin monomers[11]. Increasing concentrations of yeast profilin accelerated exchange of ATP to ATP-ATTO-488 (and inversely of ATP-ATTO-488 to ATP) by 3- to 4-fold on actin monomers (Fig. 4a and Supplementary Fig. 4a), similar to what was previously published for ATP[12]. This observation is consistent with the location of the binding site of profilin on actin, which does not overlap with the predicted position of ATTO-488 in the structure (Fig. 4b). We also labeled profilin with KU560, a long fluorescence lifetime ($\tau_{fl} \approx 20$ ns) dye, to quantify this interaction by fluorescence anisotropy. Our results revealed similar affinities of KU560-profilin for G-actin bound to ATP or ATP-ATTO-488 (Fig. 4c and Supplementary Table 2). We then worked with ADF/cofilin, which inhibits nucleotide exchange on actin[29]. Similar to the results with ATP, the exchange of ATP to ATP-ATTO-488 (and inversely of ATP-ATTO-488 to ATP) on actin monomers was inhibited by ADF/cofilin (Fig. 4d and Supplementary Fig. 4a). Also this result was consistent with the binding site of ADF/cofilin, which does not overlap with the predicted position of ATTO-488 (Fig. 4e). Moreover, we measured a similar affinity of a KU560-labeled ADF/cofilin for G-actin bound to ATP or ATP-ATTO-488 (Fig. 4f and Supplementary Table 2). Finally, we focused on an actin filament side-binding protein. Our previous work showed that some filament side-binding proteins, such as tropomyosins, inconveniently do not copolymerize well with actin when a fraction of actin subunits are covalently labeled to fluorophores such as Alexa-568[14]. On the contrary, the use of a fluorescent ATP (ATP-Cy5) did not prevent copolymerization of GFP-labeled tropomyosin with actin, and thus allowed simultaneous visualization of tropomyosin and actin filaments (Fig. 4g). From a structural point of view this result was somewhat surprising, because there is a potential steric clash between tropomyosin[30] and fluorescent dye on the side of an actin filament (Fig. 4h). However, based on our structural data (Supplementary Fig. 1) we hypothesize that the presence of flexible linker enables the fluorophore to place itself at different positions on each side of the tropomyosin, and thus allows tropomyosin to bind actin filaments labeled with a fluorescent ATP.

To reveal whether ATP-ATTO-488 can functionally replace ATP in actin dynamics, we determined if the fluorescent nucleotides are hydrolyzed within actin filaments. This was achieved by measuring phosphate release from actin filaments in the presence of profilin and ADF/cofilin, and in the presence or in the absence of ATP or ATP-ATTO-488. This experiment revealed a similar release of phosphate with ATP or ATP-ATTO-488, demonstrating that ATP-ATTO-488 is hydrolyzed within actin filaments (Fig. 4i and Supplementary Fig. 4b). Together, these data demonstrate that fluorescent nucleotides can be applied for labeling actin filaments without perturbing essential interactions with other ABPs, and that ATP-ATTO-488 filaments display similar rate of phosphate release compared to ATP-actin filament. Thus, ATP-ATTO-488 can be used as a robust tool to visualize actin filament dynamics in various functional experiments in the presence of ABPs.

**A biomimetic assay reveals that fluorescent nucleotides can power actin polymerization-based force generation**. We ultimately sought to test the functionality of this family of fluorescent nucleotides with actin. One of the main mechanisms of producing work from ATP hydrolysis is through actin polymerization itself, which generates protrusive forces when actin filament barbed ends elongate against a load[2–4]. Bead motility assays[10,18] are standard in vitro experiments where ATP provides energy for actin-based propulsion (Fig. 5a). In these assays, the progressive disassembly of pre-polymerized actin filaments serves as a source of actin monomers (Fig. 5a – Step 1). As disassembling subunits are bound to ADP, exchange for new ATP molecules (Fig. 5a – Step 2) is required so that actin monomers can re-polymerize productively at the surface of polystyrene microbeads (Fig. 5a – Step 3).

We used a bead-motility assay with purified proteins. Beads were functionalized with WASp, which is a nucleation promoting factor of the Arp2/3 complex. WASp activates the Arp2/3 complex at the surface of the beads, and hence triggers the nucleation of new actin filaments and the formation of propulsive branched actin networks. In the absence of ATP in the system, we observed little to no actin assembly and beads remained non-motile (Fig. 5b). Titration with increasing concentrations of ATP progressively restored the formation of actin comet tails, which propelled the beads. The effect was progressive for concentrations of ATP lower than 100 μM, and then plateaued (Supplementary Fig. 5a). Interestingly, even at low concentration of ATP, the rate of the beads remained constant for long periods of time (at least up to 30 min) (Supplementary Fig. 5b). This result indicates that the motility of the beads is likely to be limited by the rate of nucleotide exchange rather than by a rapid depletion of the ATP pool. Substitution of ATP for ATP-ATTO-488 maintained the formation of actin tails and bead movement (Fig. 5b), proving unambiguously that ATP-ATTO-488 is functional for actin-based force generation. However, we found that beads moved slower for equivalent concentrations of ATP-ATTO-488, indicating that although functional, ATP-ATTO-488 is slightly less efficient than ATP (Fig. 5b and Supplementary Fig. 5b). This is most likely due to slightly slower dissociation rate of ATP-ATTO-488 from actin monomers compared to the dissociation rate of ATP (Fig. 2).

We also tested $N^6$-(6-Amino)hexyl-ATPs conjugated to various other dyes (Fig. 5c–e). All ATP analogs of this family tested here could bind to actin and integrate within actin tails when ATP was additionally present (Fig. 5d). However, we found differences in their ability to fuel actin-based motility in the absence of ATP (Fig. 5c). Importantly, when using 10% of labeled-ATP and 90% of unlabeled ATP in the bead motility

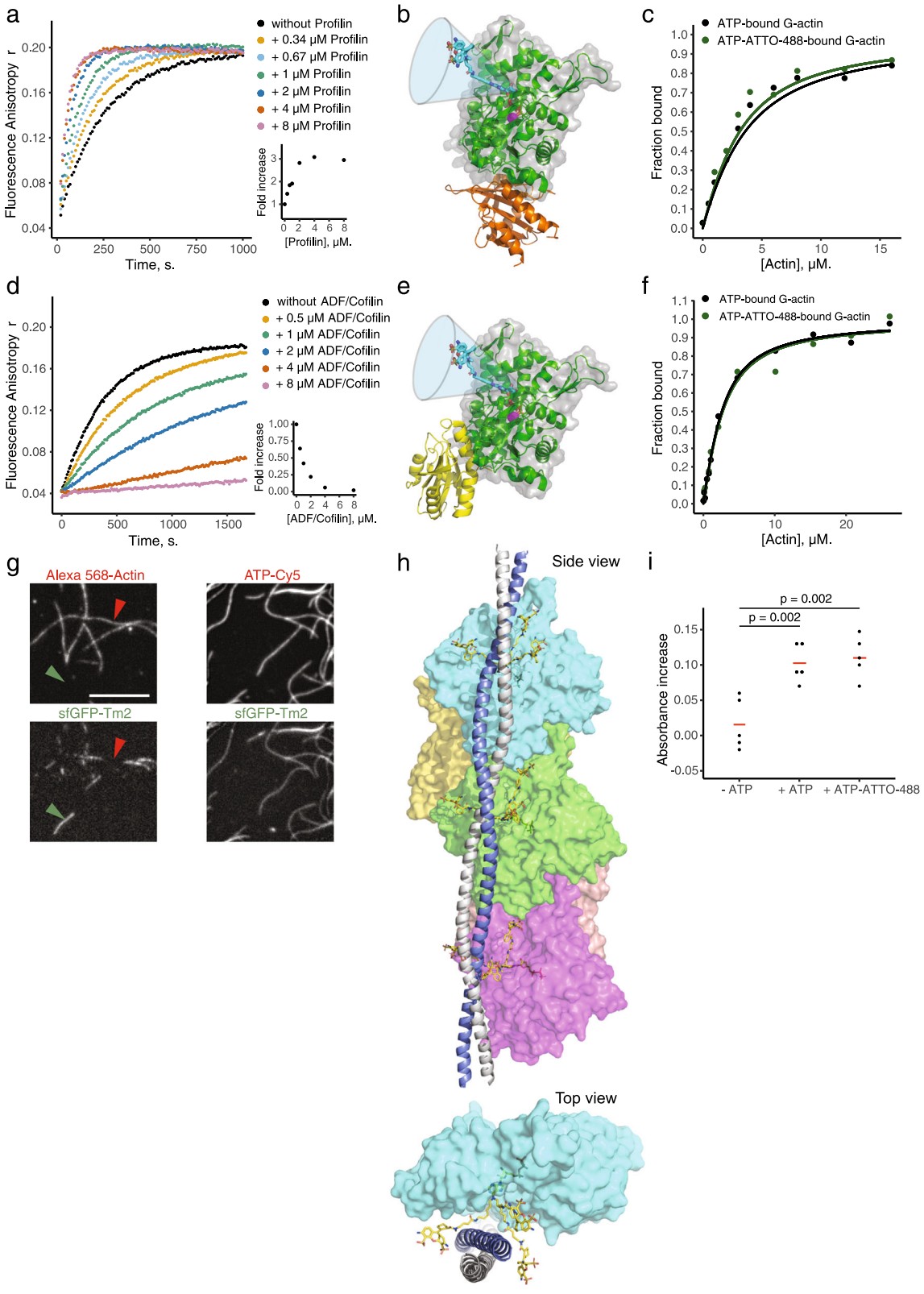

assay, all ATP analogs of this family could drive actin-based motility with indistinguishable speed compared to control conditions in the presence of unlabeled ATP only (Fig. 5e). Thus, when used in sub-stoichiometric ratios to unlabeled ATP, all these molecules can be applied to label actin filaments and networks without affecting actin-based force production. Moreover, our results identify a selection of four fluorescent ATPs,

ranging over a large band of the visible spectrum, which can be used in these assays without ATP: ATP-ATTO-488, ATP-ATTO-532, ATP-Cy3, and ATP-Cy5 (Supplementary Fig. 5c).

## Discussion
This study identifies a family of fluorescent nucleotides, $N^6$-(6-Amino)hexyl-ATP derivatives, which binds to actin. In-depth

**Fig. 4 ATP-ATTO-488 does not prevent the binding or activity of various essential actin-binding proteins, and is hydrolyzed within actin filaments.**
ATP adenosine triphosphate, ADF actin-depolymerizing factor. **a** Binding kinetics of ATP-ATTO-488 (0.2 µM) to G-actin (2 µM) in the presence of increasing concentration of profilin. Experiments were performed in NFG + MEI buffer. Inset is a quantification of the rate of increase at initial time points as a function of profilin's concentration. **b** Illustration of relative position of the ATP-ATTO-488 (cyan, stick representation) with its multiple conformations (funnel) and profilin (orange), when both are bound to an actin monomer (green). **c** Fluorescence anisotropy assay for the binding of KU560-profilin to G-actin bound to ATP (black) or ATP-ATTO-488 (in green). 5-fold excess of ATP-ATTO-488 (or ATP for the control condition) was pre-incubated with G-actin for 150 min in G-buffer at room temperature to reach 88% occupancy of ATP-ATTO-488. 0.5 µM KU560-profilin was titrated with increasing concentrations of G-actin in G-buffer. Fit curves indicate $K_d = 2.1 \pm 0.5$ µM for G-ATP actin and $2.6 \pm 0.6$ µM for G-ATP-ATTO-488 actin. **d** Binding kinetics of ATP-ATTO 488 (0.2 µM) to G-actin (2 µM) in the presence of increasing concentration of ADF/cofilin. Experiments were performed in NFG + MEI buffer. Inset is a quantification of the rate change at initial time points as a function of ADF/cofilin's concentration. **e** Illustration of the relative position of ATP-ATTO-488 (cyan, stick representation) with its multiple conformations (funnel) and ADF/cofilin (yellow) when both are bound to actin (green).
**f** Fluorescence anisotropy assay for the binding of KU560-ADF/cofilin to G-actin bound to ATP (black) or ATP-ATTO-488 (in green). 5-fold excess of ATP-ATTO-488 (or ATP for the control condition) was pre-incubated with G-actin for 150 min in G-buffer at room temperature to reach 88% occupancy of ATP-ATTO-488. In all, 2 µM KU560-ADF/cofilin was titrated with increasing concentrations of G-actin in G-buffer. Fit curves indicate $K_d = 1.7 \pm 0.5$ µM for G-ATP actin and $1.6 \pm 0.3$ µM for G-ATP-ATTO-488 actin. **g** Scale bar: 10 µm. Left: TIRFM images of 200 nM tropomyosin sfGFP-Tpm-1.6 copolymerized with Alexa-568-labeled actin filaments (0.8 µM; 10% labeled), inducing a partial segregation of sfGFP-Tpm-1.6 -rich (green arrowhead) and Alexa-568-rich (red arrowhead) sections of actin filaments (Pearson's coefficient $r = 0.497$). Right: TIRFM images of 200 nM tropomyosin sfGFP-Tpm-1.6 copolymerized with unlabeled actin (0.8 µM) and ATP-Cy5 (0.4 µM), showing a better colocalization of the two probes (Pearson's coefficient $r = 0.795$). At similar G-actin and ATP-Cy5 concentrations, steady-state anisotropy after nucleotide exchange predicts a 49% occupancy of ATP-Cy5 ($r_{MIN} = 0.242 \pm 0.001$; $r_{EQ} = 0.268 \pm 0.001$; $r_{MAX} = 0.295 \pm 0.001$), which is consistent with the homogenous fluorescence signal of ATP-Cy5 along actin filaments. Images were taken 15–20 min after the initiation of the experiment. This experiment was repeated independently three times. **h** Visual representation of the theoretically possible positions of ATP-ATTO-488 when superimposed on the F-actin/ tropomyosin structure. There are multiple ways by which the ATTO-488 dye (in yellow), linked to ATP (green) can be positioned without physically interfering with the α-helical tropomyosin molecules. **i** Phosphate release from actin filaments measured 30 min after incubation of F-actin (25 µM), profilin (1.5 µM), ADF/cofilin (1.5 µM), MESG (0.2 mM), PNP (2 units), and in the absence of in the presence of ATP or ATP-ATTO-488 (33 µM). Red bars indicate average values. $n = 5$. Statistical significance is given with p-values from one-factor ANOVA tests. Source data are provided as a Source Data file.

characterization of this interaction demonstrates that these probes do not interfere with actin polymerization, and do not prevent the interactions of ABPs such as profilin, ADF/cofilin or tropomyosin with actin. Future experiments using fluorescent ATP derivatives with actin and other ABPs, especially ABPs binding proximally to the ATP-binding pocket of actin, should confirm beforehand whether normal interactions are preserved or not. The N[6]-(6-Amino)hexyl-ATP derivatives are also hydrolyzed within actin filaments and can serve as functional energy sources in actin polymerization-based force generation. Thus, this family of fluorescent nucleotides represents a versatile class of actin probes for studying actin dynamics and functions of ABPs. These fluorescent nucleotides also provide a robust tool for labeling divergent actins, and perhaps also actin-related proteins, for various biochemical and in vitro imaging experiments.

**Advantages and limitations.** Fluorescent nucleotides offer many advantages for studying actin energetics. Firstly, these ATP derivatives are available commercially as adenosine triphosphates (ATP) as well as adenosine diphosphates (ADP) with a large selection of fluorescent moieties. Secondly, some of these ATP derivatives, including ATP-ATTO-488, are conjugated to fluorescent probes whose fluorescence lifetimes (typically several ns) are well adapted to fluorescence anisotropy. We have demonstrated that recording variations of fluorescent anisotropies over time is a powerful method for quantitatively measuring the binding and unbinding kinetics of these molecules to actin. Thirdly, these superior fluorescent dyes, which have been developed specifically for microscopy, are highly sensitive and enable imaging down to the single molecule level. By fluorescence anisotropy, we demonstrated that it is possible to precisely detect their binding or dissociation from actin at concentrations as low as 20 pM, which is a 4 order-of-magnitude improvement over previous probes. This sensitivity enables studying nucleotide exchange on actin monomers under physiological conditions, at concentrations below the critical concentration of actin assembly. Fourthly, fluorescent nucleotides offer a much more robust method for visualizing actin assembly as

compared to chemical labeling of actin or fluorescent actin probes. Labeling does not involve any specific modification of actin. The presence of a flexible linker allows the fluorophore to change its position when interactions with ABPs are proximal to the ATP binding pocket. We have shown that individual actin filaments can be imaged by TIRFM, that actin networks can be imaged by epi-fluorescence, and it is expected that this labeling works well for actins from different species.

Although advantageous, a number of precautions need to be taken with these methods. Firstly, because the fluorescent probe is bound to the base of the ATP and not to the phosphate groups, one has to remember that it allows tracking nucleotide exchange on actin, but not ATP hydrolysis within actin filaments. Secondly, the presence of the fluorophore has a small impact on nucleotide binding and unbinding rates. Particularly, our results indicate that ATP-ATTO-488 dissociates from G-actin about three times slower than ATP. This difference reveals a slightly higher affinity of ATP-ATTO-488 for actin than ATP, which explains the slower exchange rate. We speculate that a slower exchange rate accounts for a slower movement of the beads in the presence of ATP-ATTO-488, but it is also possible that other biochemical reactions could be mildly affected by the presence of ATP-ATTO-488 in this biomimetic experiment. Thirdly, only a subset of these ATP derivatives are functional as energy sources for actin-based motility. We could not generalize any rule from the limited number of fluorophores tested in this study, but we suspect that functionality is limited to ATPs conjugated to the smallest and non-charged fluorophores. Nevertheless, when used in sub-stoichiometric ratios to unlabeled ATP, all ATP derivatives of this family can be applied to label actin filament networks without affecting actin-based force production. Overall, the use for specific experiments of dye conjugates other than ATP-ATTO-488 for specific experiments would first require their full characterization as done in this study.

**Tools and methods to study actin dynamics and energetics.** The methods described in this study allow a powerful way of

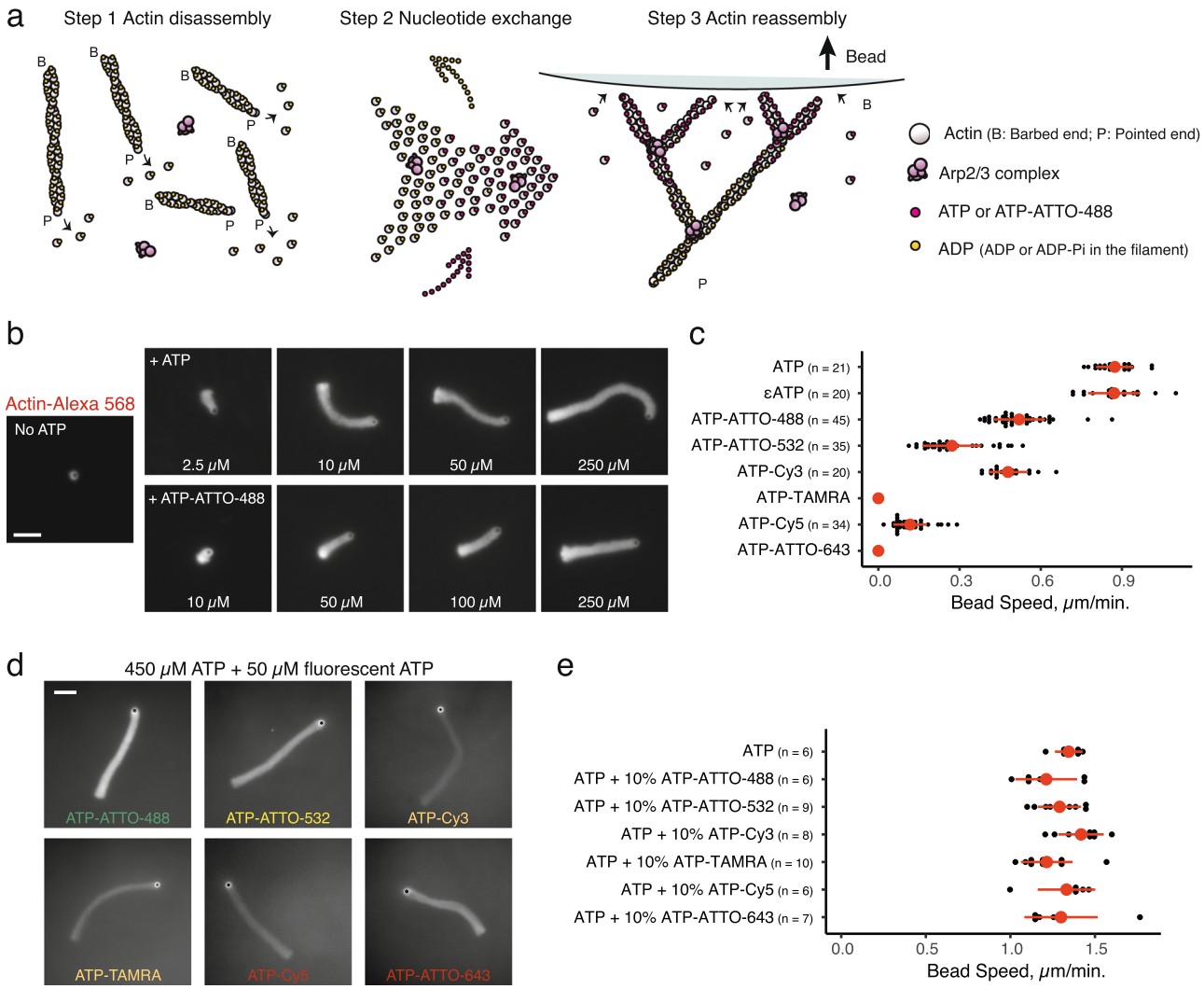

**Fig. 5 Design of a biomimetic assay to evaluate the potency of fluorescent ATPs as energy sources for actin-based force generation.** ATP adenosine triphosphate, Arp actin-related protein, TAMRA tetramethylrhodamine. Scale bars: 10 μm. **a** Rational design of experiments performed from pre-assembled unlabeled actin filaments to investigate the functionality of fluorescent ATPs in an actin-based motility assay. Fluorescent ATPs are added in solution so that they can replace ADP from disassembling actin monomers, fuel the bead motility assay, and integrate within branched actin networks. **b** Fluorescence snapshots of branched-actin comets propelling 2-μm diameter WASp-coated microbeads in the presence of F-actin (1% Alexa-568-labeled for imaging), Arp2/3 complex, profilin, ADF/cofilin, capping protein, and variable concentration of unlabeled ATP (upper images) or ATP-ATTO-488 (lower images), 30 min after initiation of the experiment, showing that ATP-ATTO-488 can fuel a bead motility assay. Red letters are to clarify that the fluorescence signal of Actin-Alexa-568 is recorded in this experiment. This experiment was repeated independently three times. **c** Bead velocity in assays performed in the presence of $N^6$-(6-Amino)hexyl-ATP fluorescent derivatives (50 μM) conjugated to a selection of fluorophores, showing that not all fluorescent ATPs are valid energy sources. Red dots indicate average values and red bars standard deviations. **d** Fluorescence snapshots of branched-actin comets in the presence of ATP (450 μM) and fluorescent ATP (50 μM), 30 min after initiation of the experiment. Colored letters are to clarify that the fluorescence signal of ATP derivatives are recorded in this experiment. This experiment was repeated independently three times. **e** Beads velocity in the presence of ATP (450 μM) and fluorescent ATP (50 μM), showing that although not all fluorescent ATPs are functional, they can still be used as markers of actin network assembly. Red dots indicate average values and red bars standard deviations. Source data are provided as a Source Data file.

quantifying the dynamics of the nucleotide bound state during actin network formation and turnover. They offer a way to track the integration and dissociation of ATP and ADP-G-actin within single actin filaments or biologically-relevant actin networks. They also offer a means to dissect the subtleties of the mechanisms related to actin recycling, and understand in more detail the contribution of factors involved in the regulation of actin dynamics, such as those involved in actin network disassembly or those catalyzing nucleotide exchange. Moreover, these probes will enable us to investigate in detail the principles related to consumption of energy by actin networks.

## Methods

**Protein purification and labeling**. Rabbit muscle actin was purified from muscle acetone powder as described in[31], and stored at 4 °C in G buffer (5 mM Tris pH 8; 0.1 mM $CaCl_2$; 0.2 mM ATP; 0.5 mM DTT; 0.02% Sodium Azide). For pyrene assays, actin was labeled on Cys-374 with pyrene iodoacetamide as described in[32,33]. For microscopy assays, actin was labeled on lysines with Alexa-568-succinimidyl ester (ThermoFisher Scientific). Actin was dialyzed for 24 h in a buffer containing 20 mM Hepes pH 7.5, 50 mM KCl, 0.1 mM $CaCl_2$ and 0.2 mM ATP, and then labeled with a 6-fold excess of fluorophore overnight at 4 °C. Labeled actin filaments were pelleted at 390,000×g for 40 min. The supernatant was eliminated and actin filaments from the pellet were resuspended and dialyzed in G-buffer overnight at 4 °C. Insoluble fraction was pelleted by another centrifugation at 390,000×g for 40 min, and actin monomers in the supernatant

were eventually loaded on a Sephadex G-25 (Sigma-Aldrich) column to eliminate the remaining free fluorophore.

The following proteins were overexpressed in Rosetta 2(DE3)pLysS cells and purified as described in the following references: *S. cerevisiae* profilin (Pfy1p)[34], *S. cerevisiae* WASp (Gst-Las17(375-Cter)-6xHis)[35], *S. cerevisiae* ADF/cofilin (Cof1p)[36], and *S. cerevisiae* capping protein (Cap1p/Cap2p)[35]. Human tropomyosin-1.6 (Tpm1.6) N-terminally fused to sfGFP (sfGFP-Tpm1.6) was purified as described in Gateva et al.[14]. Native Arp2/3 complex was purified from *S. cerevisiae* as described in Antkowiak et al.[35]. For Pfy1p labeling, we designed a S36C/C89A mutant based on previous mutations tested on *Acanthamoeba*'s profilin[11]. For Cof1p labeling, we used the previously published D34C Cof1p construct[37]. Both constructs were sub-cloned in a pRSFDuet1 vector for protein expression with a 6xHis N-terminal affinity purification tag and a TEV-cleavage site. Protein expression was induced in Rosetta 2(DE3)pLysS cells with 1 mM IPTG overnight at 20 °C. Bacteria were lysed in 10 mM Tris–HCl pH 7.5, 1 mM TCEP, 150 mM NaCl, 10 mM Imidazole pH 7.0, 0.1% Triton X-100, 5% glycerol and protease inhibitors (Complete Protease Inhibitor Cocktail, Roche). After clearing the bacterial lysate by centrifugation at 160,000×*g* for 20 min, proteins were batch purified on Nickel-Sepharose beads 6 Fast Flow (GE Healthcare Life Sciences, Piscataway, NJ, USA). Protein concentration was estimated on SDS-page gel and proteins were labeled by addition of a 5-fold excess of KU560 rigid maleimide dye (KU dyes) overnight at 4 °C. Unbound fluorophore was eliminated by additional batch purification and labeled proteins were eluted by TEV-cleavage from Nickel-Sepharose for 1 h at room temperature. Proteins were concentrated with an Amicon Ultra 4 ml device (Merck4Biosciences) and dialyzed against storage buffer (10 mM Tris pH 8; 150 mM NaCl; 2 mM DTT; 5% glycerol).

**Fluorescent nucleotides analogs**. Fluorescent nucleotides used in this study were purchased from Jena Bioscience (https://www.jenabioscience.com/). They include N[6]-(6-Amino)hexyl-ATP-ATTO-488 (ref. NU-805-488), N[6]-(6-Amino)hexyl-ATP-ATTO-532 (ref. NU-805-532), N[6]-(6-Amino)hexyl-ATP-Cy3 (ref. NU-805-CY3), N[6]-(6-Amino)hexyl-ATP-5/6-TAMRA (ref. NU-805-TAM), N[6]-(6-Amino)hexyl-ATP-ATTO-643 (ref. NU-805-643), N[6]-(6-Amino)hexyl-ATP-Cy5 (ref. NU-805-CY5), EDA-ATP-ATTO-488 (ref. NU-808-488), γ-[6-Aminohexyl]-ATP-ATTO-488 (ref. NU-833-488), and 1,N[6]-etheno-ATP (εATP; ref. NU-1103). Fluorescent nucleotide analogs were aliquoted upon reception and stored at −80 °C, so that experiments could be performed from samples which have undergone <2–3 freeze/thaw cycles. When required, fluorescent nucleotide analogs were diluted in 20 mM Hepes pH 7.5.

**Fluorescence anisotropy kinetics**. Sample preparation: for nucleotide exchange experiments in the presence of actin monomers, G-actin was diluted first from its stock concentration to a concentration of 100 μM, and then to the final concentration of experiment in a nucleotide-free G-buffer (NFG buffer; 5 mM Tris pH 8; 0.2 mM CaCl₂; 0.5 mM DTT; 0.02% Sodium Azide). This dilution lowers the concentration of free ATP to favor the exchange of a high fraction of fluorescent nucleotide, and allows to control the remaining free ATP concentration for each experiment. Therefore, experiments initiated in this work from G-actin are performed in the presence of a two-fold excess of ATP over G-actin, unless otherwise stated. Fluorescent nucleotide analogs were added at the indicated concentration to initiate nucleotide exchange. Experiments were performed in NFG buffer (for exchange on calcium-bound G-actin), in NFG buffer supplemented with 10x MEI (NFG + MEI buffer; 1x MEI containing 1 mM MgCl₂, 1 mM EGTA, 10 mM Imidazole-HCl, pH 7.0; for exchange on magnesium-bound G-actin), or NFG buffer supplemented with 10x KMEI (NFG + KMEI buffer; 1x KMEI containing 50 mM KCl, 1 mM MgCl₂, 1 mM EGTA, 10 mM Imidazole-HCl, pH 7.0; for exchange on magnesium-bound actin in polymerizing conditions).

Data recording: anisotropy values were acquired on a Safas Xenius XC spectrofluorimeter (Safas Monaco). Values of anisotropy are given based on a standard formula:

$$r = \frac{I_{VV} - G \times I_{VH}}{I_{VV} + 2 \times G \times I_{VH}} \quad (1)$$

where *V* and *H* represent the positions of the polarizer and analyzer and *G* is the grating factor:

$$G = \frac{I_{HV}}{I_{HH}} \quad (2)$$

We requested a specific fluorescence anisotropy kinetic acquisition mode, which was developed by SAFAS and is available from the version 7.8.13.0 of the SP2000 software. Signals for ATTO-488 (resp. ATP-Cy5) were acquired by excitation at 504 nm (resp. 651 nm) and emission at 521 nm (resp. 671 nm).

**Nucleotide exchange with etheno-ATP**. For an appropriate comparison of the two fluorescent probes, εATP was mixed to G-actin in the same conditions than for the experiments performed with ATP-ATTO-488. The εATP fluorescence signal was acquired by an excitation at 338 nm and a detection at 410 nm. Acquisition of the εATP and ATP-ATTO-488 signals were performed on the same Safas Xenius XC spectrofluorimeter, and the acquisition settings (exposure time and photomultiplier tension) were optimized to collect a similar signal for each sample.

**Conversion of anisotropy values into concentrations of bound and unbound ATP-ATTO-488**. Extreme values of anisotropy, corresponding to conditions were 0% or 100% of ATP-ATTO-488 is bound to actin, were evaluated. The minimal value of anisotropy ($r_{MIN}$) corresponded to the anisotropy factor *r* measured when only free nucleotides were present in solution. The maximal value of anisotropy ($r_{MAX}$) was evaluated by two different methods. Firstly, we took into consideration the fact that the fluorescence lifetime of 4.1 ns of ATTO-488 is short enough so that values of anisotropy do not change if the probe is bound to any protein larger than 40 kDa[38]. Therefore, we measured this value for ATP-ATTO-488 bound to filamentous actin (F-actin). We pre-incubated G-actin (10 μM) with ATP-ATTO-488 (1 μM) for 30 min at room temperature in 1x MEI. Polymerization was started subsequently with the addition of 50 mM KCl for 1 h. F-actin bound to a fraction of ATP-ATTO-488 was sedimented by centrifugation at 265,000×*g* for 2 h. Supernatant containing the remaining free ATP-ATTO-488 molecules was removed, the pellet containing F-actin was resuspended in an equivalent volume of 1x KMEI, and the value of anisotropy was immediately measured. Secondly, because the mobility of ATP-ATTO-488 could be slightly different when bound to F- or G-actin, we incubated G-actin (2 μM) with ATP-ATTO-488 (0.2 μM) for 4 h at room temperature in NFG buffer. Sample was then incubated with Dowex 1×8 ion exchange resin for 2 h at 4 °C to remove free nucleotides in solution (Sigma-Aldrich)[39]. Dowex resin was then eliminated by low-speed centrifugation and anisotropy was immediately measured. $r_{MAX}$ values measured with both methods were in good agreement, suggesting that mobility of ATTO-488 is not very different whether ATP-ATTO-488 is bound to F-actin or G-actin. $r_{MAX}$ values measured with the second method were on average only 7% lower than with the first method, but it is difficult to conclude whether this difference is due to the presence of more free ATP-ATTO-488 left in solution with the second method, to differences in molecular weights between F- and G-actin, or to slight differences of ATP-ATTO-488's mobility when bound to F- versus G-actin. This 7% difference between the two methods does not lead to any major variation in the final kinetic parameters in our model, and $r_{MAX}$ values indicated in this work were always determined with the first method. From the two extreme values of anisotropy, a linear scale of binding was derived for all intermediate anisotropy values, which was converted into concentrations of bound ATP-ATTO-488 by multiplying the initial concentration of ATP-ATTO-488 added in solution.

**Modeling and determination of kinetic parameters**. Kinetic parameters and standard deviations were determined using Kintek Explorer version 1. We used a simple model based on the following chemical equation:

$$G - Actin - ATP = G - Actin + ATP \quad (3)$$

$$G - Actin + ATP - ATTO - 488 = G - Actin - ATP - ATTO - 488 \quad (4)$$

All curves corresponding to the association of ATP-ATTO-488 and to the dissociation of ATP-ATTO-488 in the presence of an excess of ATP were fitted with the same set of parameters. This model was then used to estimate the percentage of ATP-ATTO-488 occupancy in experiments where ATP-ATTO-488 was pre-incubated with G-actin.

**Kinetics of pyrene-actin assembly**. Unlabeled actin and pyrene-actin were mixed to reach a final percentage of pyrene labeling of 5%. Actin polymerization was initiated by addition of 1X KMEI. The pyrene fluorescence signal was acquired by an excitation at 365 nm and a detection at 407 nm. Polymerization kinetics were acquired on the same Safas Xenius XC spectrofluorimeter.

**Steady-state fluorescence anisotropy with KU560-labeled profilin and ADF/cofilin**. G-actin was incubated in NFG buffer for 150 min at room temperature with a 5-fold excess of ATP-ATTO-488, in order to reach a final percentage labeling of 88%, or with a similar amount of ATP. KU-560-Pfy1 (0.5 μM) and KU560-Cof1 (2 μM) were incubated with variable concentration of G-actin, and steady-state anisotropy values were recorded by exciting KU560 at 555 nm and collecting light at 650 nm. Affinity constants were determined with R version 3.4.3 from nonlinear least-square estimates (nls function) of the parameters of the following equation[40]:

$$r \sim r_{\min} + (r_{\max} - r_{\min}) \left\{ \frac{K_d + [G] + [ABP] - \sqrt{(K_d + [G] + [ABP])^2 - 4 \cdot [G] \cdot [ABP]}}{2 \cdot [ABP]} \right\} \quad (5)$$

where *r* is the measured anisotropy, $r_{\min}$ is the anisotropy value of unbound KU560-Pfy1 or KU-560-Cof1, $r_{\max}$ is the anisotropy value of KU560-Pfy1 or KU-560-Cof1 when bound to G-actin, [G] is the total concentration of G-actin and [ABP] is the total concentration of KU560-Pfy1 or KU-560-Cof1. Fractions bound were determined from the estimated values of $r_{\min}$ and $r_{\max}$.

**Structure determination of the G-actin/ADF-H domain complex in the presence of ATP-ATTO-488**. For crystallization, the C-terminal ADF-H domain of twinfilin (residues 176–316) was expressed and purified from *E. coli* as a SUMO-fusion protein as described in Kotila et al.[41], and rabbit muscle actin was purchased

from Cytoskeleton (Akl-99). One milligram of actin was diluted into 400 μl of milliQ water and incubated on ice for 1 h, and dialyzed against 2 liters of dialysis buffer (5 mM HEPES, 0.1 mM MgCl$_2$, 0.05 mM EGTA, 0.1 mM ADP (96% purity, Sigma-Aldrich A5763), 0.5 mM β-Mercaptoethanol, pH 8.0) by using a 0.5 ml Slide-A-lyzer mini device (10 kDa cutoff, Pierce) at +4 °C for 16 h. The solution was ultracentrifuged for 20 min at 435,000 × g by using a TLA-100 rotor, and the concentration of actin was determined against the dialysis buffer. Subsequently, ATP-ATTO-488, CaCl$_2$ and Tris-HCl were added to solution to reach final concentrations of 0.2 mM, 0.25 mM, and 10 mM (pH 7.5), respectively. After 2.5-hour incubation on ice, 1.1 molar excess of twinfilin was added to the actin solution, followed by mixing incubation for 5 min. NaCl was the added to a final concentration of 50 mM, and the protein complex was concentrated to 6–10 mg/ml by using a 3-kDa cutoff Amicon Ultra Centrifugal Filter Device. The obtained protein complex was used immediately for crystallization trials.

The crystallization was carried out by mixing 100 nl of protein solution with 100 nl of mother solution in a sitting drop setup at 20 °C. Visible rod-like crystals appeared after 6 h in 0.1 M sodium cacodylate (pH 6.0) and 15% (w/v) polyethylene glycol.

For remote collection of diffraction data, the crystals were cryo-protected by soaking in LV CryoOil (MiTeGen) and snap-frozen in N$_2$ for shipping. The data were collected at 100 K using 0.9686 Å wavelength, Pilatus3 6 M detector, 10% transmission power, 0.05 s exposure, and 0.1° oscillation angle as a total of 2000 frames at Diamond Light Source (UK, Didcot) beamline I24. We observed some radiation damage in the late frames, therefore only 1650 first frames were used in data reduction. The data had several lattices but the major lattice contained 76% of all reflections and was used for indexing with X-ray Detector Software (XDS). The data were merged and scaled with AIMLESS (CCP4) to the 2.56 Å resolution utilizing autoPROC toolbox (Supplementary Table 1). The initial molecular replacement solution was obtained with Phaser by using 3DAW as a search model. Rounds of manual model building in COOT and refinement with BUSTER lead to final model with $R_{work}$ = 0.17 and $R_{free}$ = 0.23. We also considered anisotropic treatment of the data using ellipsoidal cut (STARANISO), which gained some additional resolution down to 2.145 Å, but with lower completeness. We used anisotropically treated data to polish and finalize the final model, especially in the placement of waters and some side chains, as well as to confirm that we do have some additional density, that we have modeled partially with water molecules but might represent the partial occupancy by the ATTO-488 dye, conjugated to the ATP nucleotide. In addition, during the refinement process we have observed negative density on the gamma phosphate, indicating that some of the molecules are in ADP rather than ATP form. Adjusting the occupancy of the gamma phosphorus atom and 3 connected oxygen atoms to 0.7, solved the problem. Thus, we propose that the presence of a small fraction (~30%) of ADP in the nucleotide-binding pocket of actin in crystals is due to either from the presence of remaining residual amount of ADP in actin buffer after the nucleotide exchange protocol described above, or from very slow hydrolysis and Pi release of the ATP-ATTO-488 in actin/ADF-H domain complex.

ATP-ATTO-488 was manually docked to a rabbit muscle actin monomer structure (3DAW, chain A) by overlaying its ATP moiety with the ATP molecule in the actin monomer crystal structure. The structures of the profilin-actin complex (2BTF), and tropomyosin-decorated F-actin (5JLF) are from[30,42]. Cofilin-actin monomer complex is represented by the structure of an actin monomer in complex with the C-terminal ADF-H domain of mouse twinfilin-1 (3DAW, chain B)[28].

**Single molecule and single actin filament imaging**. For single molecule experiments, glass slide, and coverslips were cleaned by a plasma exposure for 3 min at 80–90 W (Harrick Plasma) and stored up to 1 week at room temperature. For the single filament experiments, slides and coverslips were incubated after plasma treatment with 1 mg/ml Silane-PEG 5 K in a solution of ethanol complemented with 0.1% HCl for 18 h at room temperature under gentle shaking, then washed extensively in ethanol and water, dried, and stored at 4 °C for up to 1 week.

For the single molecule experiments, Alexa-568-labeled actin monomers were incubated with ATP-ATTO-488 in MEI buffer for 15 min at room temperature to reach maximal binding. Actin monomers were then diluted in imaging buffer 1 (10 mM imidazole-HCl, pH 7.0; 1 mM EGTA; 1 mM MgCl$_2$; 70 mM DTT; 2.5 mg/ml glucose; 15 μg/ml catalase; 70 μg/ml glucose oxidase and 0.3% methylcellulose 1500 cP), deposited between a clean slide and coverslip and imaged immediately by TIRFM.

For actin polymerization experiments, unlabeled and Alexa-568-labeled G-actins were mixed to reach a final labeling ratio of 7.5%. Polymerization was initiated by diluting actin, profilin, and ATP-ATTO-488 in imaging buffer 2 (10 mM imidazole-HCl, pH 7.0; 50 mM KCl; 1 mM EGTA; 1 mM MgCl$_2$; 70 mM DTT; 2.5 mg/ml glucose; 15 μg/ml catalase; 70 μg/ml glucose oxidase; 0.1% BSA and 0.3% methylcellulose 1500 cP). A 3-μl sample was deposited between Si-PEG passivated slides and coverslips and imaged by TIRFM. Like for bulk assays, remaining free ATP concentration is two-fold over the concentration of G-actin in these experiments.

TIRFM assays were performed on a Nikon Eclipse Ti microscope, equipped with a ×60 NA 1,49 objective, an OptoSplit II beam splitter, a Prime 95B scientific CMOS camera (Photometrics), and using Metamorph software version 7.10.1.161.

**Actin-based motility assays**. Unlabeled and Alexa-568-labeled G-actins were mixed to obtain a final labeling percentage of 1%. Actin was pre-polymerized overnight at 4 °C in NFG + KMEI buffer, to eliminate a maximum amount of free ATP. Actin network assembly at the surface of Las17-coated beads was initiated by incubating F-actin and other proteins in imaging buffer 3 (20 mM Hepes pH 7.5; 100 mM KCl; 2 mM MgCl$_2$; 50 mM DTT; 0.3 mg/ml glucose; 0.03 mg/ml catalase; 0.15 mg/ml glucose oxidase; 0.8% methylcellulose 1500 cP; 0.5% BSA and the indicated concentration of ATP. Optimal protein concentrations of 8 μM F-actin, 1 μM profilin, 0.5 μM ADF/cofilin, 1 μM capping protein, and 250 nM Arp2/3 complex were used for all experiments.

Images were acquired on a Zeiss Axio Observer Z1 microscope equipped with a 100x/1.4NA Oil Ph3 Plan-Apochromat objective and a Hamamatsu ORCA-Flash 4.0LT camera. Images were acquired with Zen 2.3 blue edition.

**Phosphate release from ATP hydrolysis**. ATP and ATP-ATTO-488 hydrolysis within F-actin was determined by detecting the release of inorganic phosphate with 2-amino-6-mercapto-7-methylpurine ribonucleoside (MESG) and purine-nucleoside phosphorylase (PNP) (Enzcheck, Thermoscientific). We first mixed G-Actin with profilin and ADF/cofilin and dialyzed the protein mixture overnight at 4 °C in NFG + KMEI buffer. The purpose of this dialysis was to polymerize actin under fast turnover conditions so that most of the free ATP would be hydrolyzed and removed from the filamentous solution. As ATP-ATTO-488 is expensive and sold at low concentration (1 mM), this step is necessary to subsequently detect any phosphate release in our assay. After dialysis, F-actin, profilin, and ADF/cofilin were incubated at room temperature in NFG + KMEI buffer with MESG, PNP in the presence or in the absence of ATP or ATP-ATTO-488. Phosphate release was detected over time and by comparing the absorbance of five independent samples at 360 nm after incubation and 30 min later.

**Quantification and statistical analysis**. All experiments presented in this manuscript were performed independently at least three times. We present results from one set of experiments, so that results do not include additional statistical variability that might arise from the repetition of the experiments. Numbers n indicate the number of measurements performed for each condition. We provide the average values from these measurements, and error bars represent standard deviations. Images were analyzed quantitatively with ImageJ version 1.49 v. Single actin filaments were detected with the plugin JFilament and Pearson's coefficient were calculated with the plugin JaCoP. Data were quantified and statistically analyzed with R version 3.4.3, and plotted using the package ggplot2 (https://ggplot2.tidyverse.org/). Statistical significance are given as p-values of one-factor ANOVA tests.

**Reporting summary**. Further information on experimental design is available in the Nature Research Reporting Summary linked to this paper.

## Data availability
Data supporting the findings of this manuscript are available from the corresponding author upon reasonable request. A reporting summary for this Article is available as a Supplementary Information file. The structure of G-actin/ADF-H domain complex in the presence of ATP-ATTO-488 is deposited in Protein Data Bank PDB 6YP9 [https://doi.org/10.2210/pdb6YP9/pdb]. Source data are provided with this paper.

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

## Acknowledgements
The authors acknowledge Thomas Le Goff, Natalie Dye, Benoît Dehapiot, the Centuri multi-engineering platform for their advices, and the Helsinki Crystallization facility for technical assistance. This project has received funding from the European Research Council (ERC) under the European Union's Horizon 2020 research and innovation programme (grant agreement no 638376/Segregactin) and from the Labex INFORM (ANR-11-LABX-0054, funded by the 'Investissements d'Avenir French Government program') to A.M. and from Academy of Finland (320161) to P.L.

## Author contributions
J.C. performed the spectrofluorimetric assays and analyzed these data. A.A. performed the bead motility assays and analyzed these data. K.K., T.K., and P.L. performed crystallization and structural analysis of fluorescent nucleotide binding to actin. J.E. performed a TIRFM assay. A.G. expressed, purified and labeled most of the proteins used in this work. A.M. conceived the project, designed and coordinated the experiments with all authors, performed some of the spectrofluorimetric and TIRFM assays with J.C. and J.E., analyzed the TIRFM results, and wrote the manuscript with input from all authors.

## Competing interests
The authors declare no competing interests.
