## [Peer Review File · Nature Communications]

Reviewer #1 (Remarks to the Author):

Colombo and coworkers test a number of commercially available ATP analogs conjugated to bright fluorescent dyes in different actin assembly assays in vitro. They confirm that substitutions/additions at the N6 position of the adenine base of ATP are somewhat tolerated concerning most aspects of actin assembly, while sugar and phosphate modifications are not. This is generally known in the field for more than four decades (ever since N6-Etheno-ATP was found to bind to actin with nearly unperturbed affinity and kinetics, see Thames et al 1974). The paper contains numerous conceptual and experimental issues that would require improvement (see below). Its central weakness, which cannot be addressed in a straightforward manner, however, is its purely technical nature and the lack of novel biological or biochemical insight. As such, it will be only of limited interest to a handful of biochemical actin labs, which is why I cannot recommend publication in journal aiming at a general audience such as Nature Communications.

Conceptual points:

- The emphasis on “structure-based strategies” in the identification of nucleotide analogs, which is prominently displayed in the title/abstract, is an exaggeration. No structural methods or considerations were used in the “discovery” process. The authors merely tested several commercially available ATP analogs. All structural work contained in the paper was performed after confirming N6-labeled ATP as a candidate. The authors need to tone down these misleading claims throughout.
- Similarly, promoting base-labeled ATP as “revolutionary tools” to monitor “dynamics of the nucleotide bound state” of actin is simply misleading. The authors must be aware that the main interest in the nucleotide state of actin concerns the not the presence/absence of the nucleotide as a whole, but selectively of the gamma phosphate. The ATP analogs used here cannot report on changes at the gamma phosphate position, greatly limiting their use. The authors should be much more open and clear in this regard from the beginning.
- The point of using nucleotide analogs to investigate actin polymerization “energetics” as stated in the title is not well developed. No direct measurements of “energy”-related quantities (such as force, rate of ATP consumption etc.) were performed.

Experimental points:

- The conclusion that “ATP-ATTO-488 does not affect the polymerization rate of actin filaments” (Figure 4) is not sufficiently supported by the data. The authors detect no effects at sub-stoichiometric incorporation (about 25%). Similar behavior has been observed previously for actin labeled at Cys374, which retains wt polymerization activity at least for some dyes when used at low stoichiometries. However, this does not exclude drastic effects at higher occupancy. I suggest that these experiments should be carried out with actin monomers that contain stoichiometric amounts of labeled nucleotide.
- The claim that labeled nucleotides do not affect the interaction between monomers and proteins such as profilin and cofilin (Figure 5A-D) is not supported by the data. The assays do not report on the interaction between these proteins and fluorescent ATP-containing actin monomers, because the reactions are initiated from monomers bound to unlabeled nucleotide. The authors should use monomers containing bound fluorescent ATP analogs at stoichiometric amounts at the beginning of the experiment. They should furthermore quantify the affinity between profilin/cofilin and actin

monomers and compare activities to controls containing non-labeled ATP.

- The same hold true for the conclusion that “use of fluorescent ATP did not prevent copolymerization of tropomyosin with actin”. These experiments need to be conducted at saturating amounts of Atto488-ATP and need to be quantified and compared to controls with unlabeled ATP, which are currently lacking.

- The conclusion that “ATP-ATTO-488 filaments display similar rate of phosphate release compared to ATP-actin filament” is again not sufficiently supported by the data. The data shown (Figure 5G) seems to be from an endpoint experiment that does not report on reaction kinetics.

Minor points:

- The biophysical characterization of the interaction between actin and the ATP analogs is spread over two figures (2 and 3). For the purpose of clarity, the data should be consolidated into a single figure. Given their importance, both affinities and rate constants should also be summarized in a table.

Reviewer #2 (Remarks to the Author):

This is an interesting study that describes the use of N6-(6-amino)hexyl-ATP derivatives as nucleotide analogs to study actin. It compares these derivatives with other popular derivatives, specifically N6-etheno-ATP, and concludes that they are superior for nucleotide exchange experiments and for visualization of actin monomers and filaments in vitro. The suitability of these derivatives is supported by several key experiments: 1) time-resolved fluorescence anisotropy to assess nucleotide exchange on actin, 2) total internal reflection fluorescence microscopy to monitor the binding of individual fluorescent ATP molecules to actin monomers and filaments (I found this to be the most interesting and original part of the work), 3) demonstration that these ATP derivatives do not interfere with the binding of some popular actin-binding proteins (ADF/cofilin, profilin, tropomyosin), and 4) a bead-motility assay to show that actin with bound N6-(6-amino)hexyl-ATP derivatives can generate protrusive forces. The characterizations are carefully conducted and will probably inspire researchers to use these derivatives in future studies. Therefore, I recommend publication in NC, and my comments below can be addressed without the need for additional experimental data.

The one thing I found somewhat annoying about this paper is some misleading statements in the Title, Abstract, Introduction, and Discussion that confused me (and probably will confuse other readers). First, it made me think that the authors had discovered these ATP derivatives. Indeed, the word “discover” is in the Title, and they even talk about “revolutionary tool” in the Introduction. In reality these derivatives have existed for a long time and are commercially available, and were designed for this use. This study simply makes a strong case for their use, specifically with actin. Such exaggerations are unnecessary, because the paper is interesting and valid without it. Similarly, the Title, Abstract and Introduction suggest that these derivatives were discovered based on structural considerations, but the one structure presented does not show the ATTO-488 dye and linker (even if interesting). Clearly, the N6 position of the nucleotide faces the solvent, and a dye at this location would be less disruptive to nucleotide binding than elsewhere. Therefore, I suggest to de-emphasize the “A structural approach to discover ATPase-binding fluorescent nucleotides”, as stated in the

Title, and focus on what makes this paper interesting, namely a demonstration that N6-(6-amino)hexyl-ATP derivatives are good candidates to study actin, and superior to other popular dyes. Minor points:

1. Please, include a diagram of etheno-ATP (possibly in Fig. 1a).
2. Would be important in ruling out other probes based on structural considerations and clashes (Fig. 1c) to contrast with a model of etheno-ATP (which binds actin)
3. Part of the structural data in Supp. Fig. 1 is more important than the negative data in Fig. 1b. Consider sending the former to the main text and the latter to SI.
4. Fig. 2 can be merged with Fig. 1, and the topic is related (particularly if etheno-ATP is more prominently addressed in Fig. 1.
5. Does Fig. 3d really need such a long time-axis? Probably parts b-d fit nicely on one row.

Roberto Dominguez

Reviewer #3 (Remarks to the Author):

Summary :

The authors identify fluorescent nucleotide analogs that can serve as spectroscopic probes with sensitivity for studying actin assembly and interaction with regulatory proteins, potentially to the single molecule level. The authors show that these nucleotides do not significantly disrupt native actin function and offer new opportunities to (re)evaluate fundamental aspects of actin assembly. The work will be highly appreciated by labs studying actin dynamics, particularly those working with purified protein components. However, I have some comments that I hope the authors will find useful.

Major comments

1. In the fluorescence anisotropy measurements shown in Fig. 2, 3, 5, S2 and S3, does the fluorescence total intensity change with time? If it does, it must be accounted for.
2. Is the discrepancy between the two different fluorescently labeled nucleotide binding traces trivial? A ~3-fold change?
3. On a related note, how many exponentials do the time traces follow?
4. From the x-axis of Fig. 2b and legend of Fig S2a, the lowest ATP-ATTO-488 concentration is 200 pM. Therefore, is the claimed detection limit of 20 pM of fluorescently labeled ATP accurate?
5. Some units presented in figure legends don't match that in figure captions or in the main text.
6. The dye in labeled fluorescent ATP is Cy5, which differs from those being characterized in Fig. 5e. This is confusing.
7. Is the assumption that there is no difference in the anisotropy of ATP-ATTO-488 bound to G- or F-actin valid?

8. Fig 4C – multiple filaments with and without 488-ATP analog should be measured and quantified to show the rate of growth is unaffected by the 488-ATP analog. A single picture is not as convincing.

9. Fig 5E – there is only 1 image to support the conclusion that sfGFP-Tm2 binds well to ATP-Cy5.

10. Does the dye on actin affect the incorporation of the 488-ATP analog? The authors could assemble filaments with 488-ATP analog with using 568-G-Actin and unlabeled G-Actin, then quantify the fluorescence of 488-ATP analog over at least ~50 filaments and plot as a histogram or box and whiskers plot. The experiments presented in Fig. 4 are getting there.

11. Similarly, does the 488-ATP analog affect cofilin binding?

12. Consider adding the following in the limitations section of discussion

a. For the general readers note the following, all data is only relevant when low ratios of dyes are used.

b. Only a small subset of ABPs have been tested and it should not be assumed that other ABPs are unaffected.

c. Only 1 dye (488-Atto-ATP) was fully investigated and the other dye conjugates on ATP may have issues.

Minor comments

1. Please update the exact method to label 568-actin. Although the authors refer to a previous paper, it is essential to have it in here for readers.

2. The authors need to report errors on parameter values. For example, the dissociation rate constants of labeled and unlabeled ATP, etc..

3. In Fig. 6, the units on plots, figure caption and text are different.

4. The minimum and maximum anisotropy values should be reported.

5. The units don't match in supplementary Fig. 4a and 4b. In c and d, the yellow color curves are hard to see.

6. Fig. 6b and d the units in legends are different from that in figure caption and text.

7. In fig. 3d, what is concentration for all nucleotides before and during pyrene actin polymerization?

8. What is ATP concentration in Fig. 4a and 4b?

9. Line 616, NFG buffer is actin G-buffer and it should not polymerize actin.

10. No buffer condition and experimental detail are given in Fig. 5.

11. At the end of line 846 – some text is missing?

12. In Fig. 5g there should be control for phosphate contamination for ATP and labeled ATP. The free phosphate in 33 uM labeled or unlabeled ATP could be comparable from those released in filaments.

13. Fig. 6b should make clear that for the ATP sample, pyrene was excited.

14. In Fig. 3b and 3c, the anisotropy value at two blue arrows don't match.

15. In Fig, 3a, is something wrong with bound $ATP \cdot X[ATP^*]$?

E-

Response to reviewers comments

(Manuscript NCOMMS-20-18534-T – Colombo, Antkowiak et al.)

We would like to thank all three Reviewers for their work on our manuscript and for their thoughtful comments. We have taken into account all their comments and suggestions, and sincerely believe that these changes significantly improved our manuscript. We hope that the three reviewers find the revised manuscript improved, and suitable for sharing with the scientific community.

Please find below a copy of their comments in black, and our responses in blue.

Reviewer #1 (Remarks to the Author):

Colombo and coworkers test a number of commercially available ATP analogs conjugated to bright fluorescent dyes in different actin assembly assays in vitro. They confirm that substitutions/additions at the N6 position of the adenine base of ATP are somewhat tolerated concerning most aspects of actin assembly, while sugar and phosphate modifications are not. This is generally known in the field for more than four decades (ever since N6-Etheno-ATP was found to bind to actin with nearly unperturbed affinity and kinetics, see Thames et al 1974). The paper contains numerous conceptual and experimental issues that would require improvement (see below). Its central weakness, which cannot be addressed in a straightforward manner, however, is its purely technical nature and the lack of novel biological or biochemical insight. As such, it will be only of limited interest to a handful of biochemical actin labs, which is why I cannot recommend publication in journal aiming at a general audience such as Nature Communications.

While Reviewers 2 and 3 found this study very important, and recommended publication following relatively minor revisions, the Reviewer 1 was somewhat more critical especially concerning the general importance of this method. However, we are convinced of the importance of these new tools, as many collaborators and colleagues to whom we have presented these tools/approaches have already expressed their interest in using them for their experiments. Moreover, we have successfully used this approach for studying an actin from a flagellated protozoan parasite (Kotila et al., unpublished). Such divergent actin would be difficult to label with 'standard' approaches, and the nucleotide exchange kinetics of this actin was not possible to study using etheno-ATP. This demonstrates that the new tools described in our manuscript present a robust new approach for studying also divergent actins from evolutionarily distant organisms (e.g. unicellular parasites and perhaps also archaea). We have now edited the manuscript text to better highlight the usefulness of these nucleotide analogues, and related methods, for studying a wide range of actins (and most likely also actin-related proteins).

Moreover, Reviewer 1 provided excellent suggestions for new experiments, and we thank him/her for that. We have now performed these experiments and analyses, and we believe that all these results confirm and strengthen our claims.

Conceptual points:

- The emphasis on "structure-based strategies" in the identification of nucleotide analogs, which is prominently displayed in the title/abstract, is an exaggeration. No structural methods or considerations were used in the "discovery" process. The authors merely tested several commercially available ATP analogs. All structural work contained in the paper was performed after confirming N6-labeled ATP as a candidate. The authors need to tone down these misleading claims throughout.

We accordingly modified the abstract, the discussion and changed the title of our manuscript.

- Similarly, promoting base-labeled ATP as "revolutionary tools" to monitor "dynamics of the nucleotide bound state" of actin is simply misleading. The authors must be aware that the main interest in the nucleotide state of actin concerns the not the presence/absence of the nucleotide as a whole, but selectively of the gamma phosphate. The ATP analogs used here cannot report on changes at the gamma phosphate position, greatly limiting their use. The authors should be much more open and clear in this regard from the beginning.

We agree that tracking presence/absence of gamma-phosphate would also be important. We did not claim that N6-labeled ATPs could do that, and have now added a sentence in the discussion to make this point unambiguous (lines 447-449). Nevertheless, we would like to point out that having a sensitive tool, suitable for fluorescence imaging, to follow binding and exchange of nucleotides on actin, is in our opinion very important to the cytoskeleton community.

- The point of using nucleotide analogs to investigate actin polymerization "energetics" as stated in the title is not well developed. No direct measurements of "energy"-related quantities (such as force, rate of ATP consumption etc.) were performed.

Importantly, we demonstrate in the manuscript that these molecules provide energy for actin-based motility, and thus in the future these molecules can be used in studies on actin energetics. We aim at addressing in the future important questions on force and rates of ATP consumption, but such studies are beyond the scope of this 'Method' paper. We discuss this point at the end of the manuscript (lines 498-507).

Experimental points:

- The conclusion that "ATP-ATTO-488 does not affect the polymerization rate of actin filaments" (Figure 4) is not sufficiently supported by the data. The authors detect no effects at sub-stoichiometric incorporation (about 25%). Similar behavior has been observed previously for actin labeled at Cys374, which retains wt polymerization activity at least for some dyes when used at low stoichiometries. However, this does not exclude drastic effects at higher occupancy. I suggest that these experiments should be carried out with actin monomers that contain stoichiometric amounts of labeled nucleotide.

This is an excellent suggestion, and we have performed additional TIRF experiments at higher occupancy (39% and 95%) (new Figure 3d). We do not find any significant difference in elongation rates of actin filaments, indicating that ATP-ATTO-488 does not change kinetics of actin polymerization.

- The claim that labeled nucleotides do not affect the interaction between monomers and proteins such as profilin and cofilin (Figure 5A-D) is not supported by the data. The assays do not report on the interaction between these proteins and fluorescent ATP-containing actin monomers, because the reactions are initiated from monomers bound to unlabeled nucleotide. The authors should use monomers containing bound fluorescent ATP analogs at stoichiometric amounts at the beginning of the experiment. They should furthermore quantify the affinity between profilin/cofilin and actin monomers and compare activities to controls containing non-labeled ATP.

This is also an excellent suggestion. As suggested by the Reviewer, we performed an experiment where excess ATP was added to actin monomers bound to ATP-ATTO-488 and ADF/cofilin or profilin (new Supplementary Figure 4a). These experiments show similar kinetics of ATP-ATTO-488 dissociation from actin monomers, proving unambiguously this point.

As requested, we also performed steady-state fluorescence anisotropy experiments with ADF/cofilin or profilin labeled with KU560 and increasing amount of G-actin bound to ATP or ATP-ATTO-488 (new Figures 4c and 4f). These results enable us to evaluate the affinities of ADF/cofilin and profilin to G-actin-ATP and G-actin-ATP-ATTO-488. Again, we find similar affinities whether actin is bound to ATP or ATP-ATTO-488.

- The same hold true for the conclusion that "use of fluorescent ATP did not prevent copolymerization of tropomyosin with actin". These experiments need to be conducted at saturating amounts of Atto488-ATP and need to be quantified and compared to controls with unlabeled ATP, which are currently lacking.

We provide quantification of the percentage of fluorescent ATP occupancy, and indicate that this percentage is higher than percentage of Alexa-568 labeling in the control experiment. We also re-phrased this section to avoid any controversy, because we cannot exclude that continuous binding of tropomyosin could work at very low percentage of Alexa-568 labeling. We indicate that in our experiment, at 49 % of fluorescent ATP occupancy, ATP and tropomyosin signals colocalize well (Pearson's coefficient of 0.795), whereas in the case at 10 % of Alexa 568 labeling, we see much weaker co-localization (Pearson's coefficient of 0.497). These results show that using a fluorescent ATP is more appropriate than using labeled actin in experiments where actin filaments are decorated with tropomyosin.

- The conclusion that "ATP-ATTO-488 filaments display similar rate of phosphate release compared to ATP-actin filament" is again not sufficiently supported by the data. The data shown (Figure 5G) seems to be from an endpoint experiment that does not report on reaction kinetics.

We provide now also standard time-courses of these experiments (new Supplementary Fig. 4b).

Phosphate release in these experiments was reported linear over 500 s in similar published experiments (Kotila et al. Nature Communications, 2018). Our idea with endpoint experiments is to show results from multiple experiments, as we observed some variability in these experiments.

Minor points:

- The biophysical characterization of the interaction between actin and the ATP analogs is spread over two figures (2 and 3). For the purpose of clarity, the data should be consolidated into a single figure. Given their importance, both affinities and rate constants should also be summarized in a table.

We have followed both suggestions. Figures 2 and 3 are merged and affinities and rate constants are summarized in Supplementary Table 2.

Reviewer #2 (Remarks to the Author):

This is an interesting study that describes the use of N6-(6-amino)hexyl-ATP derivatives as nucleotide analogs to study actin. It compares these derivatives with other popular derivatives, specifically N6-etheno-ATP, and concludes that they are superior for nucleotide exchange experiments and for visualization of actin monomers and filaments in vitro. The suitability of these derivatives is supported by several key experiments: 1) time-resolved fluorescence anisotropy to assess nucleotide exchange on actin, 2) total internal reflection fluorescence microscopy to monitor the binding of individual fluorescent ATP molecules to actin monomers and filaments (I found this to be the most interesting and original part of the work), 3) demonstration that these ATP derivatives do not interfere with the binding of some popular actin-binding proteins (ADF/cofilin, profilin, tropomyosin), and 4) a bead-motility assay to show that actin with bound N6-(6-amino)hexyl-ATP derivatives can generate protrusive forces. The characterizations are carefully conducted and will probably inspire researchers to use these derivatives in future studies. Therefore, I recommend publication in NC, and my comments below can be addressed without the need for additional experimental data.

The one thing I found somewhat annoying about this paper is some misleading statements in the Title, Abstract, Introduction, and Discussion that confused me (and probably will confuse other readers). First, it made me think that the authors had discovered these ATP derivatives. Indeed, the word “discover” is in the Title, and they even talk about “revolutionary tool” in the Introduction. In reality these derivatives have existed for a long time and are commercially available, and were designed for this use. This study simply makes a strong case for their use, specifically with actin. Such exaggerations are unnecessary, because the paper is interesting and valid without it. Similarly, the Title, Abstract and Introduction suggest that these derivatives were discovered based on structural considerations, but the one structure presented does not show the ATTO-488 dye and linker (even if interesting). Clearly, the N6 position of the nucleotide faces the solvent, and a dye at this location would be less disruptive to nucleotide binding than elsewhere. Therefore, I suggest to de-emphasize the “A structural approach to discover ATPase-binding fluorescent nucleotides”, as stated in the Title, and focus on what makes this paper interesting, namely a demonstration that N6-(6-amino)hexyl-ATP derivatives are good candidates to study actin, and superior to other popular dyes.

We thank Dr. Dominguez for his positive evaluation of our manuscript and for his suggestions. Following the proposed suggestions, which are also in line with reviewer #1, we have rewritten the Title, Abstract and the Introduction to better emphasize the usefulness of our findings.

Minor points:

1. Please, include a diagram of etheno-ATP (possibly in Fig. 1a).

This is done.

2. Would be important in ruling out other probes based on structural considerations and clashes (Fig. 1c) to contrast with a model of etheno-ATP (which binds actin)

The purpose of Figure 1c is to emphasize visually that other dyes, except for etheno-ATP and N6-(6-amino)hexyl-ATP, are structurally incompatible with actin. We tried to emphasize that point in the Figure Legend (lines 1008-1012)

3. Part of the structural data in Supp. Fig. 1 is more important than the negative data in Fig. 1b. Consider sending the former to the main text and the latter to SI.

We have now moved two structural data panels from Suppl. Fig. 1 to Figure 1.

4. Fig. 2 can be merged with Fig. 1, and the topic is related (particularly if etheno-ATP is more prominently addressed in Fig. 1).

Reviewer 1 suggested to combine Figures 2 and 3, and we found it very difficult to combine Figures 1, 2 and 3 together. As Figure 2 and 3 are referenced in the same paragraph of the ‘Results’ section (but not Figure 1), we followed in this case the recommendation of Reviewer 1. We hope that the Reviewer finds this acceptable.

5. Does Fig. 3d really need such a long time-axis? Probably parts b-d fit nicely on one row.

We have followed this advice and shortened the axis.

Roberto Dominguez

Reviewer #3 (Remarks to the Author):

We thank Reviewer 3 for his/her positive opinion on our manuscript and for his/her careful reading which helped us noticing two typos in the Figures.

Summary

The authors identify fluorescent nucleotide analogs that can serve as spectroscopic probes with sensitivity for studying actin assembly and interaction with regulatory proteins, potentially to the single molecule level. The authors show that these nucleotides do not significantly disrupt native actin function and offer new opportunities to (re)evaluate fundamental aspects of actin assembly. The work will be highly appreciated by labs studying actin dynamics, particularly those working with purified protein components. However, I have some comments that I hope the authors will find useful.

Major comments

1. In the fluorescence anisotropy measurements shown in Fig. 2, 3, 5, S2 and S3, does the fluorescence total intensity change with time? If it does, it must be accounted for.

The total fluorescence intensity of ATP-ATTO-488 does not change whether it is bound to G-actin or not. We performed a new experiment (New Supplementary Figure 2a) to clarify this point.

2. Is the discrepancy between the two different fluorescently labeled nucleotide binding traces trivial? A ~3-fold change?

We discuss briefly this difference in the Result section (lines 234-236). The discrepancy between the two different fluorescently labeled nucleotide binding traces is not trivial, and we speculate that this may be due to weak interactions of the fluorescent dye with actin. Many fluorescent labels/proteins affect binding and/or activity of proteins and small molecules, and this is surely something to keep in mind for future experiments.

3. On a related note, how many exponentials do the time traces follow?

Data are well-fit with monoexponentials. Small differences between data and fit are due to the fact that we fitted all curves (binding and dissociation) with the same set of kinetic parameters.

4. From the x-axis of Fig. 2b and legend of Fig S2a, the lowest ATP-ATTO-488 concentration is 200 pM. Therefore, is the claimed detection limit of 20 pM of fluorescently labeled ATP accurate?

Yes, it is accurate. Fig.2B shows detection of binding at 20 pM, although this represents a lower limit of detection which gives noisier data in our equipment.

5. Some units presented in figure legends don't match that in figure captions or in the main text.

We thank Reviewer 3 for noticing this unfortunate mistake. The mistake was for concentration of ATP required in the bead motility assays, which are obviously μM and not nM. We have corrected the Figure and Supplementary Figure.

6. The dye in labeled fluorescent ATP is Cy5, which differs from those being characterized in Fig. 5e. This is confusing.

We only had GFP-tropomyosins available in the lab, so we could not use ATP-ATTO-488 for this experiment. Therefore, and for this experiment only, we used an ATP of different color (ATP-Cy5) and we have now made this point clear in the text and in the Figure legend.

7. Is the assumption that there is no difference in the anisotropy of ATP-ATTO-488 bound to G- or F- actin valid?

This is a very interesting question. We originally reported that "The maximal value of anisotropy was difficult to evaluate with G-actin, because the presence of ATP in solution systematically kept a fraction of ATP-ATTO-488 free in solution", but we were also bothered by the possibility that the mobility of ATTO-488 would be different when bound to F-actin vs. G-actin.

We tried different protocols to solve this issue, and realized that incubation with a Dowex matrix was quite efficient at binding most of the free nucleotides left (as in De La Cruz and Pollard, *Biochemistry*, 1995), and was therefore likely to keep most of ATP-ATTO-488 bound to G-actin. So we exchanged ATP for ATP-ATTO-488 on G-actin in G-buffer for 4 h, and removed unbound nucleotides by incubation of the sample with Dowex beads.

With this protocol, we found convincing results, with r_{MAX} values on average only 7% lower than with the F-actin protocol. However, it is difficult to tell whether this small difference is due to residual free ATP-ATTO-488 in solution with this new method, or to differences in molecular weights of F- and G-actin, or to meaningful differences of anisotropy values when ATP-ATTO-488 binds to F- or G-actin. We report this alternative measurement in the 'Methods' section of the paper.

8. Fig 4C – multiple filaments with and without 488-ATP analog should be measured and quantified to show the rate of growth is unaffected by the 488-ATP analog. A single picture is not as convincing.

This comment is complementary to Reviewer 1's first Experimental Point. We provide in the new version of the manuscript additional evidence that actin filament elongation rates are not changed when actin is bound to up to 95 % of ATP-ATTO-488 (see new Figure 3d)

9. Fig 5E – there is only 1 image to support the conclusion that sfGFP-Tm2 binds well to ATP-Cy5.

We provide in the new version of the manuscript a quantification of this experiment (Pearson's coefficients). See Legend of the new Figure 4g.

10. Does the dye on actin affect the incorporation of the 488-ATP analog? The authors could assemble filaments with 488-ATP analog with using 568-G-Actin and unlabeled G-Actin, then quantify the fluorescence of 488-ATP analog over at least ~50 filaments and plot as a histogram or box and whiskers plot. The experiments presented in Fig. 4 are getting there.

As suggested by Reviewer 3, we have quantified the ATP-ATTO-488 fluorescence of filaments assembled in the absence or in the presence of 10% Alexa 568-labeled actin filaments. This piece of data appears in a new Supplementary Figure 3. We did not detect significant difference, suggesting that presence of the dye does not prevent incorporation of ATP-ATTO-488.

11. Similarly, does the 488-ATP analog affect cofilin binding?

This comment is also complementary to second Experimental point by Reviewer 1. We provide now experiments performed with profilin or ADF/cofilin labeled with KU560 to measure the affinities of profilin and ADF/cofilin to G-actin bound to ATP or to ATP-ATTO-488. For both proteins, we found similar values when G-actin was bound to ATP or to ATP-ATTO-488, proving that ATP-ATTO-488 does not affect profilin or ADF/cofilin binding to actin monomers.

12. Consider adding the following in the limitations section of discussion

a. For the general readers note the following, all data is only relevant when low ratios of dyes are used.

Following experiments recommended by Reviewer 1, we have performed many experiments at higher occupancy of ATP-ATTO-488. This experimental effort allows us to confirm most of our results even at higher dye ratios.

b. Only a small subset of ABPs have been tested and it should not be assumed that other ABPs are unaffected.

We now tried to make this point clear in the abstract "We demonstrate that these fluorescent nucleotides maintain functional interactions with a number of essential actin-binding proteins (ABPs)" (line 26), and in the discussion of the paper "Future experiments using fluorescent ATP derivatives with actin and other ABPs, especially ABPs binding in the proximity to the ATP-binding pocket of actin, should confirm beforehand whether normal interactions are preserved or not" (lines 435-438).

We did not bring this point as a limitation per se, because we think that one would need to identify first an ABP, which does not bind to G-actin anymore in the presence of ATP-ATTO-488.

c. Only 1 dye (488-Atto-ATP) was fully investigated and the other dye conjugates on ATP may have issues.

We clarified this point in the limitations (point 3/). We suspect from Figure 5 (bead motility assays) that other dye conjugates might have issues when used at high occupancy, and should be investigated further before doing experiments with them: "Overall, the use for specific experiments of dye conjugates other than ATP-ATTO-488 for specific experiments would first require their full characterization as done in this study." (lines 494-496). Though ATP-ATTO-488 seems superior over other conjugates, we still believe that at 10% fraction other conjugated-ATP analogs (see Fig. 5e) are useful in microscopy experiments to label filaments, and can be measured in a wide range of the spectrum if simultaneously other fluorescent signals are recorded. Further investigation is needed to identify the limitations of other conjugates.

Minor comments

1. Please update the exact method to label 568-actin. Although the authors refer to a previous paper, it is essential to have it in here for readers.

We have detailed this protocol in the new version of the manuscript. See lines 541-550.

2. The authors need to report errors on parameter values. For example, the dissociation rate constants of labeled and unlabeled ATP, etc..

This is done.

3. In Fig. 6, the units on plots, figure caption and text are different.

We have changed that (see Major point #5).

4. The minimum and maximum anisotropy values should be reported.

We have reported these values in the Legend of Figures where they are used (lines 1146-1147 and line 1240-1241).

5. The units don't match in supplementary Fig. 4a and 4b. In c and d, the yellow color curves are hard to see.

We have changed that (see also major point #5), as well as changed the color of the curves.

6. Fig. 6b and d the units in legends are different from that in figure caption and text.

This has been corrected (see major point #5)

7. In fig. 3d, what is concentration for all nucleotides before and during pyrene actin polymerization?

8. What is ATP concentration in Fig. 4a and 4b?

Concerning the minor points 7 and 8, we agree that our previous description of the experiments was not explicit enough, and did not permit easy comprehension of the final concentrations of ATP in each experiment. Therefore, in the new version of the manuscript, we provide a more detailed explanation in the 'Methods' section of how we performed the experiments to permit controlled final concentration of free ATP in all experiments.

We point out that the concentration of free ATP was taken into account for the simulations performed with Kintek Explorer. For clarity, we provide now also with the source data file all parameters that were used to fit the curves so that fits can be reproduced.

9. Line 616, NFG buffer is actin G-buffer and it should not polymerize actin.

The original manuscript indicated "in NFG buffer supplemented with 1x KMEI". We changed wording to NFG + KMEI buffer to avoid confusion and we provide the composition of this buffer, which contains 50 mM KCl.

10. No buffer condition and experimental detail are given in Fig. 5.

For experiments of Figure 5, essential experimental details and protein concentrations are now provided in the Legend. More detailed explanations of the protocols are provided in the 'Methods' section.

11. At the end of line 846 – some text is missing?

We removed a word "in"

12. In Fig. 5g there should be control for phosphate contamination for ATP and labeled ATP. The free phosphate in 33 μ M labeled or unlabeled ATP could be comparable from those released in filaments.

Reviewer 1 requested time-courses of these experiments, which eliminate the possibility that free phosphate accounts for signal increase in these experiments (if so, signal increase should be rapid and not progressive over 30 minutes). This new data are provided in the new Supplementary Figure 4b.

13. Fig. 6b should make clear that for the ATP sample, pyrene was excited.

Please note that Figure 6b represents bead motility assays, where we did not use any pyrene. The only Figure where pyrene was used was Figure 3d, where the Figure legend indicates clearly that pyrene was excited.

14. In Fig. 3b and 3c, the anisotropy value at two blue arrows don't match.

Thank you very much for noticing this mistake. We accidentally used a file from a previous analysis which was incorrect. We have corrected this mistake, and y axis values now match.

15. In Fig, 3a, is something wrong with bound $ATP \times [ATP^*]$?

We verified the formula. The concentration of fluorescent ATP bound to G-actin at time t is equal to the initial concentration of free fluorescent ATP multiplied by the percentage of fluorescent ATP bound at time t.

E-

Reviewer #1 (Remarks to the Author):

On the plus side, the authors have fixed the most important experimental issues that were pointed out. From a technical point of view, the paper is now in a state that can be accepted.

I found the textual revisions on the other hand rather lackluster, because they do not substantially change the overall misleading tone of the first version. Just as an example, I quote the first two sentences of the discussion:

“This study demonstrates that a careful structural analysis of ATPases helps predicting specific chemistries ... With this strategy, we identified a family of fluorescent nucleotides...”

Neither of these statements applies to the manuscript as already clearly stated in my first review! I could go on here, but it seems that the authors are either not willing or not able to make impactful changes to the text.

In general, I also remain sceptic concerning the conceptual advance and novel biological insight the authors achieved through this work.

Reviewer #3 (Remarks to the Author):

The authors have satisfied my concerns. I appreciate their attention to all of the concerns and comments that were raised.

E-

Response to reviewers comments
(Manuscript NCOMMS-20-18534-B – Colombo, Antkowiak et al.)

Please find below our response to the new comments of the Reviewers. Their comments appear in black, and our responses in blue.

Reviewer #1 (Remarks to the Author):

On the plus side, the authors have fixed the most important experimental issues that were pointed out. From a technical point of view, the paper is now in a state that can be accepted.

I found the textual revisions on the other hand rather lackluster, because they do not substantially change the overall misleading tone of the first version. Just as an example, I quote the first two sentences of the discussion:

“This study demonstrates that a careful structural analysis of ATPases helps predicting specific chemistries ... With this strategy, we identified a family of fluorescent nucleotides...”
Neither of these statements applies to the manuscript as already clearly stated in my first review! I could go on here, but it seems that the authors are either not willing or not able to make impactful changes to the text.

In general, I also remain sceptic concerning the conceptual advance and novel biological insight the authors achieved through this work.

We thank Reviewer 1 for appreciating that this work is technically sound and can be accepted.

We have modified the first sentences of the discussion to satisfy Reviewer 1. From now on, the manuscript mentions only a structural explanation of our results, but no longer a structural prediction.

Reviewer #3 (Remarks to the Author):

The authors have satisfied my concerns. I appreciate their attention to all of the concerns and comments that were raised.

E-

We thank Reviewer 3 for this appreciation.